# Structural and evolutionary insights into the eukaryotic RNase MRP ribonucleoprotein complex

Bin Zhou[1,2,8], Xiaozhu Wang [2,8], Futang Wan[3,8], Shaobai Li[2], Xiaoshuang Zhang[2], Yuanyuan Zhang[2], Ming Tan [1,2], Mi Cao[2], Yafeng Shen[2], Rui Gao[4], Yanjie Zhang [1] ✉, Pengfei Lan [5,6] ✉, Jian Wu [2] ✉ & Ming Lei [2,6,7] ✉

RNase MRP is a conserved eukaryotic ribonucleoprotein essential for precursor-rRNA processing and ribosome assembly. Despite previous studies of yeast RNase MRP, the composition of RNase MRP and how it adapts to process flexible, single-stranded rRNA substrates in most eukaryotes remain enigmatic. Here, we perform an integrative structural, evolutionary, and functional dissection of human RNase MRP. Using structure-based bioinformatics and cryo-EM structural analyses, we identify NEPRO (RMP64) and C18orf21 (RMP24) as the bona fide subunits unique to RNase MRP, which are indispensable for precursor-rRNA cleavage, ribosome assembly, protein synthesis, and chondrogenesis. The structure of human RNase MRP reveals a unique 'double-anchor' substrate-binding mechanism that underlies evolutionary adaptations conferring broad substrate specificity. Our work on RNase MRP provides a unified evolutionary and mechanistic framework for this essential ancient ribozyme.

Ribonuclease MRP (RNase MRP) is an evolutionarily conserved ribonucleoprotein complex that catalyzes a pivotal endonucleolytic cleavage within precursor ribosomal RNAs (pre-rRNAs), thereby facilitating the production of mature rRNAs required for ribosome assembly[1–8]. Mutations in the RNA subunit of human RNase MRP (RMRP) lead to cartilage-hair hypoplasia (CHH) with short-limbed dwarfism, sparse hair, and immunodeficiency, or to anauxetic dysplasia (AD) with severe short stature and other skeletal abnormalities, underscoring the physiological significance of this ribozyme[9–15]. Despite its broad taxonomic presence[16–20] and critical functional importance, RNase MRP has remained understudied compared to its close relative, RNase P, which employs a well-defined substrate-binding platform to process the acceptor stem of precursor transfer RNAs

(pre-tRNAs)[6,21–25]. In contrast, RNase MRP processes single-stranded RNA (ssRNA) substrates[4,26–29], posing a longstanding question of how its ribonucleoprotein framework had adapted to recognize such structurally distinct substrates.

Much of our current understanding of this question comes from the studies of budding yeast *Saccharomyces cerevisiae*, where RNase MRP shares a conserved catalytic RNA core and most protein subunits with RNase P[3,6,27,30]. Yet, it diverges through some specialized elements that support the single-stranded pre-rRNA cleavage—high-resolution structural studies of *S. cerevisiae* RNase MRP reveal a modular architecture featuring a specialized substrate-binding pocket formed by coordinated RNA motifs and protein subunits, including the RNase MRP-specific Rmp1 and the refolded Pop1 and Pop4[27,30]. These

[1]Department of Oncology, Ninth People's Hospital, Shanghai Jiao Tong University School of Medicine, Shanghai, China. [2]Shanghai Institute of Precision Medicine, Ninth People's Hospital, Shanghai Jiao Tong University School of Medicine, Shanghai, China. [3]Cancer Institute, The Affiliated Hospital of Qingdao University, Qingdao Cancer Institute, Qingdao University, Qingdao, China. [4]Frontier Science Center for Stem Cell Research, Tongji University, Shanghai, China. [5]Institute of Aging and Tissue Regeneration, Renji Hospital, Shanghai Jiao Tong University School of Medicine, Shanghai, China. [6]State Key Laboratory of Systems Medicine for Cancer, Shanghai Jiao Tong University School of Medicine, Shanghai, China. [7]Shanghai Academy of Natural Sciences (SANS), Shanghai Jiao Tong University, Shanghai, China. [8]These authors contributed equally: Bin Zhou, Xiaozhu Wang, Futang Wan. ✉e-mail: zhangyanjie@shsmu.edu.cn; pengfeilan@shsmu.edu.cn; wujian@shsmu.edu.cn; leim@shsmu.edu.cn

elements make direct contact with the single-stranded pre-rRNA substrate, conferring a substrate specificity distinct from that of RNase P[27,31]. Moreover, bioinformatics analysis has uncovered an RNA element unique to RNase MRP—stem P8 pentaloop with a 'GARAR' (R: purine) consensus—that could distinguish RNase MRP from RNase P across diverse eukaryotes from protists to metazoans[31,32]. Notwithstanding these findings, budding yeasts appear to be evolutionarily more divergent from other eukaryotes. Regular sequence-based bioinformatic searching algorithms failed to find homologs of Rmp1 and Snm1—two *S. cerevisiae* proteins unique to RNase MRP[33,34]—in other eukaryotic genomes[35]. In addition, although sharing a conserved P8 pentaloop, the RNA component of *S. cerevisiae* RNase MRP, Nme1, is markedly different from those in other eukaryotic RNase MRPs[18,19,32,36–39]. The universal molecular composition of RNase MRP and the mechanism of ssRNA substrate recognition remain enigmatic in most eukaryotes.

Here, we employ a structure-based bioinformatic approach to identify *S. cerevisiae* Rmp1 and Snm1 homologs, NEPRO and C18orf21, as the bona fide subunits of human RNase MRP, consistent with recent reports[40–42]. We also show that these proteins are indispensable for efficient pre-rRNA processing, ribosome assembly, and protein synthesis and that their depletion leads to impaired chondrogenic differentiation. Furthermore, by determining a high-resolution cryo-electron microscopy (cryo-EM) structure of the human RNase MRP holoenzyme and integrating comparative structural analyses and functional assays, we unveil an RNase MRP-unique 'double-anchor mechanism' finetuned for the recognition of flexible ssRNA substrates. This mechanism helps explain how RNase MRP diverges from RNase P in substrate recognition specificity while retaining an evolutionarily conserved catalytic core, offering a unified framework for understanding RNase MRP's unique mode of catalytic action.

## Results

### Identification of NEPRO (RMP64) and C18orf21 (RMP24) as two conserved subunits unique to RNase MRP

To decipher the molecular composition of RNase MRP, we first set out to ask whether homologs of Rmp1 and Snm1—two *S. cerevisiae* proteins that have been considered unique subunits to yeast RNase MRP—could be found in human and other eukaryotic genomes (Fig. 1a). We systematically compared the experimentally determined structural motifs of Rmp1 and Snm1 against AlphaFold predicted structural databases across diverse eukaryotic lineages (Fig. 1b and Supplementary Data 1-3). This approach designated NEPRO (nucleolus and neural progenitor protein) and C18orf21 as high-confidence structural homologs to *S. cerevisiae* Rmp1 and Snm1, respectively (Fig. 1b and Supplementary Figs. 1a, b and 2a, b). Phylogenetic analysis showed that NEPRO and C18orf21 are ancestral genes conserved in a broad range of eukaryotes from protists to metazoans (Supplementary Figs. 1b, 2b, 3, and 4). Despite low overall sequence similarities, the AlphaFold predicted models reveal compelling structural similarities, wherein NEPRO exhibits a helical bundle architecture reminiscent of *S. cerevisiae* Rmp1's tube-like structure, while C18orf21 harbors a sandwiched zinc finger motif exactly mirroring *S. cerevisiae* Snm1's spatial topology (Fig. 1b and Supplementary Figs. 1b, 2b, 3, 4). This analysis suggests that NEPRO and C18orf21 likely have been preserved through structural rather than sequence constraints during the evolution of eukaryotes (Supplementary Figs. 1b, 2b, 3, 4).

To validate whether NEPRO and C18orf21 are the genuine, specific components of human RNase MRP, we performed tandem affinity purification coupled with mass spectrometry analysis (TAP-MS) using human RNase P and/or RNase MRP subunits as baits. This assay revealed that both NEPRO and C18orf21 could be consistently co-purified with two different combinations of RNase P-MRP shared subunits, indicative of a stable association of these proteins in the RNase P or MRP complexes (Fig. 1c and Supplementary Data 4). TAP-

MS experiments using NEPRO as bait selectively enriched NEPRO and C18orf21 but not the RNase P-specific subunit RPP21 (Fig. 1d and Supplementary Data 4). Conversely, RPP21 failed to co-purify either NEPRO or C18orf21 (Fig. 1d and Supplementary Data 4). Further, parallel RNA analysis demonstrated that the complex enriched by RPP21 only contained the RPPH1 RNA of RNase P (Fig. 1e). In contrast, the complex enriched by NEPRO only contained RMRP, the RNA subunit of human RNase MRP (Fig. 1e). In addition, a previous study also showed that C18orf21 is associated with RMRP[43]. Together, these data strongly indicate that NEPRO and C18orf21 are the protein components specific to human RNase MRP.

Next, we examined whether this NEPRO-C18orf21-containing complex possesses the nuclease activity to process human pre-rRNA at the internal transcribed spacer 1 site-2 (ITS1-2) between 18S and 5.8S, a well-known cleavage site by human RNase MRP[4]. Indeed, the purified NEPRO-C18orf21-enriched complex could cleave ITS1, yet it exhibited no processing activity for pre-tRNA substrates (Fig. 1f–h). On the contrary, the RPP21-enriched complex can only process the pre-tRNA but not ITS1 (Fig. 1f–h). Therefore, these enzymatic activity analyses unambiguously affirmed that the NEPRO-C18orf21-specific complex is bona fide human RNase MRP[40–42], whereas the RPP21-specific complex is human RNase P (Fig. 1i). Hereafter, we refer to NEPRO and C18orf21 as RMP64 and RMP24, respectively (Fig. 1i).

### RMP64 and RMP24 are essential for ribosome biogenesis and protein synthesis

To investigate whether RMP64 and RMP24 are required for pre-rRNA processing in cells, we stably knocked down RMP64 or RMP24 in HEK293T cells using lentiviral small hairpin RNAs (shRNAs) targeting each gene (Supplementary Figs. 5, 6). We monitored the amount of pre-cleaved ITS1 signals in cells using fluorescence in situ hybridization (FISH) (Supplementary Fig. 5a, b)[4]. Compared with control cells, RMP64 or RMP24 knockdown cells displayed a marked increase in nuclear pre-cleaved ITS1 signals (Supplementary Figs. 5c, 6a, b). In alignment with these FISH data, quantitative RT-PCR analysis also revealed elevated pre-cleaved ITS1 levels relative to the 5'-external transcribed spacer (5'-ETS) in RMP64 or RMP24 knockdown cells (Supplementary Figs. 5d, 6c). Together, these results demonstrate that both RMP64 and RMP24 are essential components of human RNase MRP and that depletion of either leads to a defect in pre-rRNA processing in cells.

Because efficient pre-rRNA processing is integral to ribosome biogenesis, we next monitored the ribosome assembly process in RMP64 and RMP24 knockdown cells (Supplementary Figs. 5e, 6d). Sucrose gradient centrifugation analysis revealed a substantial reduction in the 40S and 80S fractions, suggestive of a defect in ribosome assembly (Supplementary Figs. 5e, 6d). Given the central role of ribosomes in protein synthesis, we then measured the incorporation of methionine analog L-homopropargylglycine (HPG) into nascent polypeptides to evaluate the effect of RMP64 or RMP24 depletion in protein synthesis. We found that both RMP64 and RMP24 knockdown cells exhibited a marked decrease in the HPG signals (Supplementary Figs. 5f, 6e). Furthermore, depletion of these subunits resulted in pronounced growth defects characterized by reduced proliferation without loss of viability (Supplementary Figs. 5g, 6f). The more severe phenotype observed upon POP1 knockdown compared with RMP64 or RMP24 depletion underscores that concurrent impairment of RNase P and MRP more strongly affects cellular growth (Supplementary Fig. 6g)[44]. These results strongly suggest that both RMP64 and RMP24 are indispensable components of the human RNase MRP holoenzyme and are essential for ribosome biogenesis, global protein synthesis, and cell proliferation.

Biallelic variations in *RMP64* have been reported in individuals with CHH and AD[45–47]. To evaluate the functional contribution of *RMP64* to chondrogenesis, we performed two-dimensional (2D) and

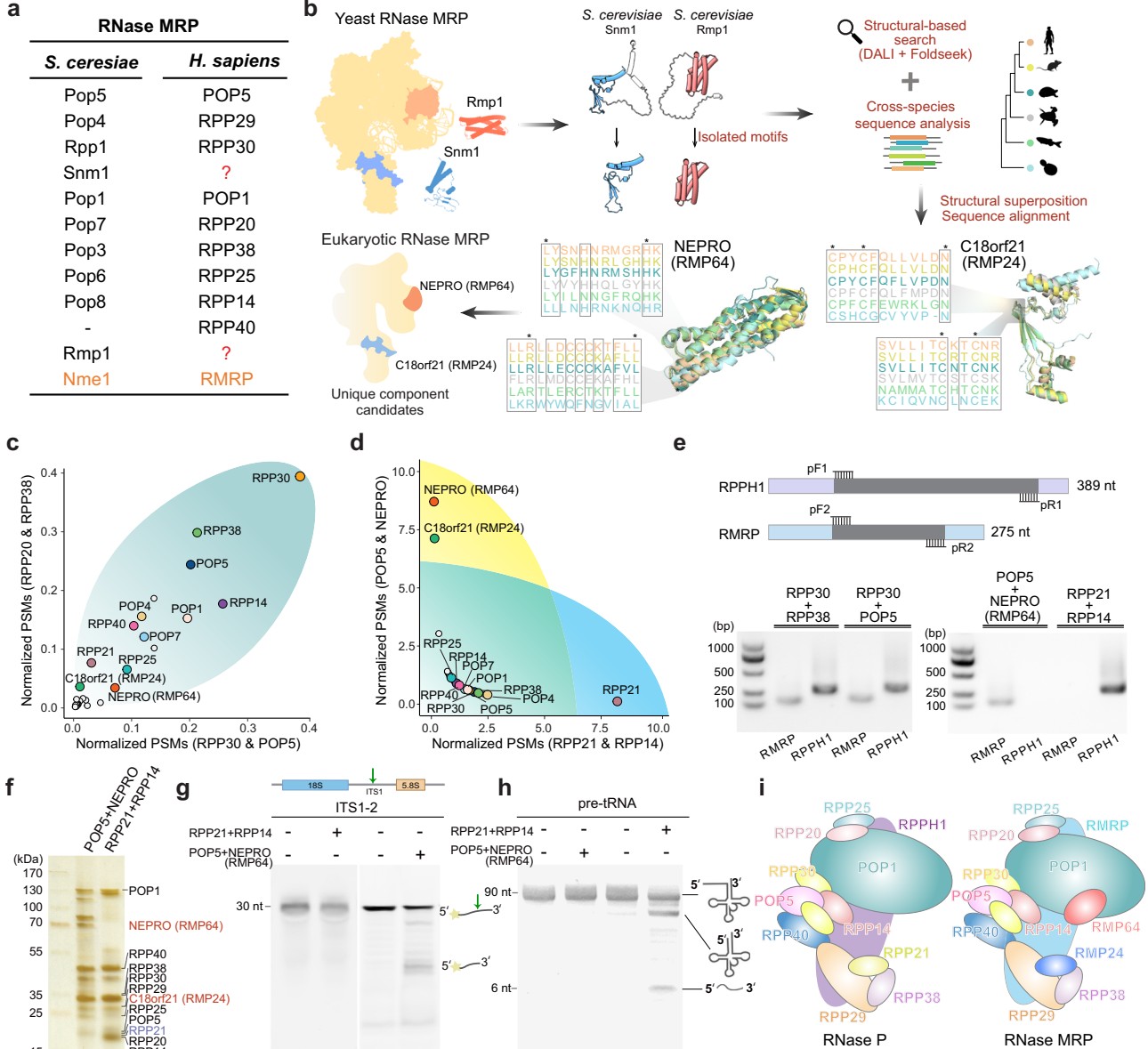

**Fig. 1 | Identification of NEPRO (RMP64) and C18orf21 (RMP24) as two conserved subunits unique to RNase MRP. a** Subunit composition of yeast and human RNase MRP. Homologous proteins are aligned. The RNA components are in orange, and red question marks indicate subunits absent in human RNase MRP. **b** Structural-based search pipeline to identify homologs of yeast Snm1 and Rmp1 across eukaryotes. Asterisks denote conserved residues. **c, d** Scatter plots showing the normalized peptide spectral matches (PSMs) of major proteins detected in the two-step affinity-purified complexes using the RNase P/MRP shared subunits (RPP30 and POP5 or RPP20 and RPP38) (**c**) or RNase P (RPP21 and RPP14) and MRP (POP5 and NEPRO) unique subunits (**d**) as the affinity baits. RNase P and MRP subunits are highlighted in various colors, while other proteins are colored in light grey. **e** Characterization of the RNA components in the purified RNase P and MRP complexes. Upper panel: The RNA components isolated from different complexes via affinity purification using distinct protein baits were reverse-transcribed into

cDNAs, which were then amplified by polymerase chain reaction (PCR) with sequence-specific primers. Lower panel: Shared RNase P and MRP protein baits co-purified both RNase P and MRP RNAs, RPPH1 and RMRP. By contrast, RNase P- and MRP-specific subunits only co-purified RPPH1 or RMRP. **f** SDS-PAGE analysis of human RNase P and MRP compositions followed by silver staining. The RNase P and MRP complexes were isolated using the RNase P- and MRP-specific proteins RPP21 and NEPRO as baits, respectively. RNase MRP-specific subunits NEPRO and C18orf21 are highlighted in red. Protein identities were assigned based on mass spectrometry data and correlated with their calculated molecular weights. **g, h** Substrate specificity of human RNase P and MRP complexes. In vitro processing assay demonstrated distinct substrate preferences; the RNase MRP complex exclusively cleaved pre-rRNA (**g**), while the RNase P complex specifically processed pre-tRNA (**h**). **i** Schematic models of human RNase P and RNase MRP complexes. Experiments shown are representative of at least three biological replicates.

three-dimensional (3D) chondrogenic induction experiments. Mouse bone marrow-derived multipotent stromal cells were induced in chondrogenic media, and the result showed that *RMP64* knockdown cells exhibited a clear defect in forming acidic polysaccharide cartilage matrix with reduced alcian blue stain (Supplementary Fig. 5h-j). Consistent with the clinical observations, these findings highlight the critical function of *RMP64* in skeletal development, especially cartilage matrix formation, and demonstrate that defects in ribosome

biogenesis caused by deficient RNase MRP can directly affect chondrogenic differentiation.

## Structure of human RNase MRP
To gain more mechanistic and evolutionary insights into RNase MRP, we purified the endogenous human RNase MRP complex and determined its cryo-EM structure at an overall resolution of ~3.5 Å (Supplementary Fig. 7 and Supplementary Table 1). 2D classification

revealed a bipartite architecture composed of a rigid, large lobe housing the catalytic center and a conformationally more dynamic small lobe (Supplementary Fig. 7a, f). Local-focused classification and refinement of the large lobe enhanced its resolution to ~2.8 Å (Supplementary Fig. 7b–e), resolving critical RNA-protein interactions within the catalytic core (Fig. 2a, b, and Supplementary Fig. 7b–e). The final structural model of human RNase MRP comprises one catalytic RNA—RMRP—and eleven distinct protein factors, including the newly identified RNase MRP-specific subunits RMP64 and RMP24 (Fig. 2a, b, and Supplementary Fig. 8). The holoenzyme adopts an intricate, asymmetrical architecture characterized by a large RMRP scaffold enwrapped by a hook-shaped network of proteins (Fig. 2b, c). The C-terminal domain of POP1 (POP1$_{CTD}$), the RPP20-RPP25 heterodimer, and the POP5-(RPP30)$_2$-RPP14-RPP40 heteropentamer wrap around RMRP in the large lobe of the complex (Fig. 2b and Supplementary Fig. 9a). Such an architecture highly resembles the organizational framework observed in human RNase P and yeast RNase P and MRP structures[27,30,48,49], underscoring the close evolutionary kinship among these eukaryotic ribozyme complexes (Supplementary Fig. 9a–d).

The RMRP RNA adopts an extended, single-layered architecture that can be divided into two domains—a catalytic (C) domain and a specificity (S) domain (Fig. 2d, e)[3,32]. The C-domain, residing in the large lobe of the complex, mainly comprises three parallel, coaxially stacked helices, stems P19–P2-P3, P1-P4, and P9-P8, in which were embedded three highly conserved regions—CR-I, CR-IV, and CR-V—that together fold into a catalytic pseudoknot motif (Fig. 2e)[3,32]. The C-domain of RMRP also features a 3-base-pair stem, P5, immediately adjacent to CR-I of the pseudoknot (Fig. 2d, e). By this partition of the RNA, three out of four stems at the C-S domain boundary (P5, P8, and P9) are all vested in the C-domain of RMRP (Fig. 2d, e), and only the long, continuous stem, P10-P12.1-P12.2, belongs to the S-domain (Fig. 2d, e). This long stem folds into a triangular-shaped conformation with a kink-turn between P12.1 and P12.2, completely different from the S-domain structure of yeast Nme1 RNA (Fig. 2e)[3,27,30].

The structure of human RNase MRP shows that RMP64 and RMP24 indeed structurally resemble their yeast counterparts, Rmp1 and Snm1, and occupy equivalent positions in human RNase MRP as Rmp1 and Snm1 do in the yeast complex (Supplementary Fig. 9a, b). In the head of the protein hook, the proximal end of RMP64 associates with the N-terminal motifs of POP1 (POP1$_{NTM}$) and RPP29 (RPP29$_{NTM}$) to form the RMP64-POP1$_{NTM}$-RPP29$_{NTM}$ ternary module, lying atop the P5 groove near the catalytic center in a conformation reminiscent of the Rmp1-Pop1$_{NTM}$-Pop4$_{NTM}$ module in S. cerevisiae RNase MRP (Fig. 2f, Supplementary Figs. 9a, b and 10a, b). Notably, similar to the yeast module, both human POP1$_{NTM}$ and RPP29$_{NTM}$ undergo a marked refolding from their conformations in RNase P into those in the RMP64-POP1$_{NTM}$-RPP29$_{NTM}$ module in RNase MRP (Supplementary Fig. 10a-d)[27,30]. Despite low sequence similarity, AlphaFold prediction showed that a RMP64-POP1$_{NTM}$-RPP29$_{NTM}$-like ternary module with refolded POP1$_{NTM}$ and RPP29$_{NTM}$ is an evolutionarily conserved structural feature in RNase MRP across diverse eukaryotic phyla from protists to metazoans (Fig. 2g and Supplementary Figs. 3 and 11a, b).

At the distal end of the protein hook, similar to S. cerevisiae Snm1, RMP24 takes the position of RPP21 in RNase P, joining with RPP29 and RPP38 to stabilize the small lobe of the RNase MRP complex (Fig. 2h, and Supplementary Fig. 12a–d). Notably, RMP24 has a large insertion in its central region, which mediates extensive electrostatic contacts with stems P10 and P12.1 and helps maintain the S-domain of RMRP in a sharply bent conformation (Fig. 2h). This insertion in RMP24 with varying lengths is another evolutionarily conserved feature of RNase MRP across diverse eukaryotic phyla (Supplementary Figs. 4 and 12e). RPP21 in RNase P, by contrast, lacks this insertion, and if present, the insertion would clash with the CR-II/III T-loop anchor in the RPPH1 RNA of human RNase P (Supplementary Fig. 12a-c). Instead, RPP21 has its uniqueness. The C-terminus of RPP21 folds back and, together with

both the N- and C-terminal extensions of RPP38, forms a unique platform to hold the CR-II/III anchor of the RNA for pre-tRNA substrate recognition (Supplementary Fig. 12c, d)[48,49]. Due to the absence of CR-II/III in RMRP, however, this platform does not exist in human RNase MRP, and all the terminal extensions of RMP24 and RPP38 are disordered and thus not visible in RNase MRP (Fig. 2h and Supplementary Fig. 12a, b). These structural incompatibilities offer a rationale for the exclusive incorporation of RMP24 and RPP21 specific into RNase MRP and RNase P, respectively. Taken together, the cryo-EM structure of human RNase MRP unveils an unequivocal structural resemblance between human and S. cerevisiae RNase MRP complexes. It demonstrates that RMP64 and RMP24 are evolutionarily conserved protein subunits unique to RNase MRP.

## Unique structural features in RNase MRP RNA

RNase P and MRP are evolutionarily closely related ribozymes. Yet, they process distinct substrates with different biophysical properties, well-folded pre-tRNAs with rigid tertiary structure versus flexible ssRNAs[4,6,28,50–52]. This prompted us to hypothesize that, accompanying the emergence of unique protein subunits RMP64 and RMP24 and the refolding mode of POP1$_{NTM}$ and RPP29$_{NTM}$, evolution might have also carved out some unique motifs in the RNA components of eukaryotic RNase P and MRP, which cooperate with their respective protein partners to play unique, indispensable roles in processing distinct substrates. The determination of human RNase MRP cryo-EM structure gave us a unique opportunity to pinpoint such structural elements. We performed a close structural comparison analysis of human RMRP with the Nme1 RNA of yeast RNase MRP as well as human and yeast RNase P RNAs, RPPH1 and Rpr1 (Fig. 3a and Supplementary Fig. 13a, b). Given the marked structural variations of the S-domains between human RMRP and yeast Nme1, this highly variable domain is unlikely to contain structural motifs conserved in all RNase MRP RNAs (Figs. 2d, e and 3a and Supplementary Fig. 13a, b). Hence, we focus our analysis on the catalytic C-domains of the RNAs.

Even though the C-domains of human and yeast RNase P and MRP RNAs share the same basic architecture, structural comparison unveiled unique features that can differentiate RNase MRP from RNase P. The most prominent yet underemphasized structural motif unique to RMRP and Nme1 is the 3-base-pair stem P5, which is immediately adjacent to the P4 stem with a sharp turn between the backbones of P4 and P5 (Fig. 3b). This unique configuration gives rise to a large, cradle-shaped surface formed by stems P4 and P5 and polynucleotides between P5 and CR-IV (Fig. 3b). On top of this surface sits the RMP64—POP1$_{NTM}$—RPP29$_{NTM}$ ternary module (Rmp1–Pop1$_{NTM}$–Pop4$_{NTM}$ in the yeast complex) (Fig. 3b). By contrast, human RPPH1 and yeast Rpr1 RNAs lack a P5 stem, but instead, they feature a P7 stem that, compared to P5, is spatially away from the catalytic pseudoknot (Fig. 3c and Supplementary Fig. 13a, b). Consequently, this forces the polynucleotides between P7 and CR-IV to take a shortcut to generate a much smaller cavity, where snugly fits the refolded POP1$_{NTM}$ motif that serves as an anchor to secure pre-tRNA in the active site of RNase P (Fig. 3c and Supplementary Fig. 10c, d)[48,49].

Second, a subtle yet consistent difference between RNase P and MRP RNAs lies in the U-shaped CR-IV (Fig. 3d, e and Supplementary Fig. 14a, b). Pairwise structural analysis unveiled that CR-IV of both human RMRP and yeast Nme1 contains six nucleotides, two shorter than the length of CR-IV in RNase P RNAs (Fig. 3d, e and Supplementary Fig. 14a, b). For comparison, hereafter, CR-IV nucleotides at equivalent positions in RNase P and MRP RNAs will be referred to as $N_i^{CR-IV}$ (i = 1,2,3,6,7,8), and the two extra nucleotides in the middle of RNase P CR-IV as $N_4^{CR-IV}$ and $N_5^{CR-IV}$ (Fig. 3d, e and Supplementary Fig. 14a, b). In both RNase P and MRP, four highly conserved nucleotides—$A_1^{CR-IV}$, $G_2^{CR-IV}$, and $A_7^{CR-IV}$, and one adenine from CR-V (A316$^{HI}$, A244$^{RMRP}$)—together weave an intricate hydrogen-bond network, stabilizing the overall U-shape of CR-IV and fixing CR-IV's relative position to the catalytic

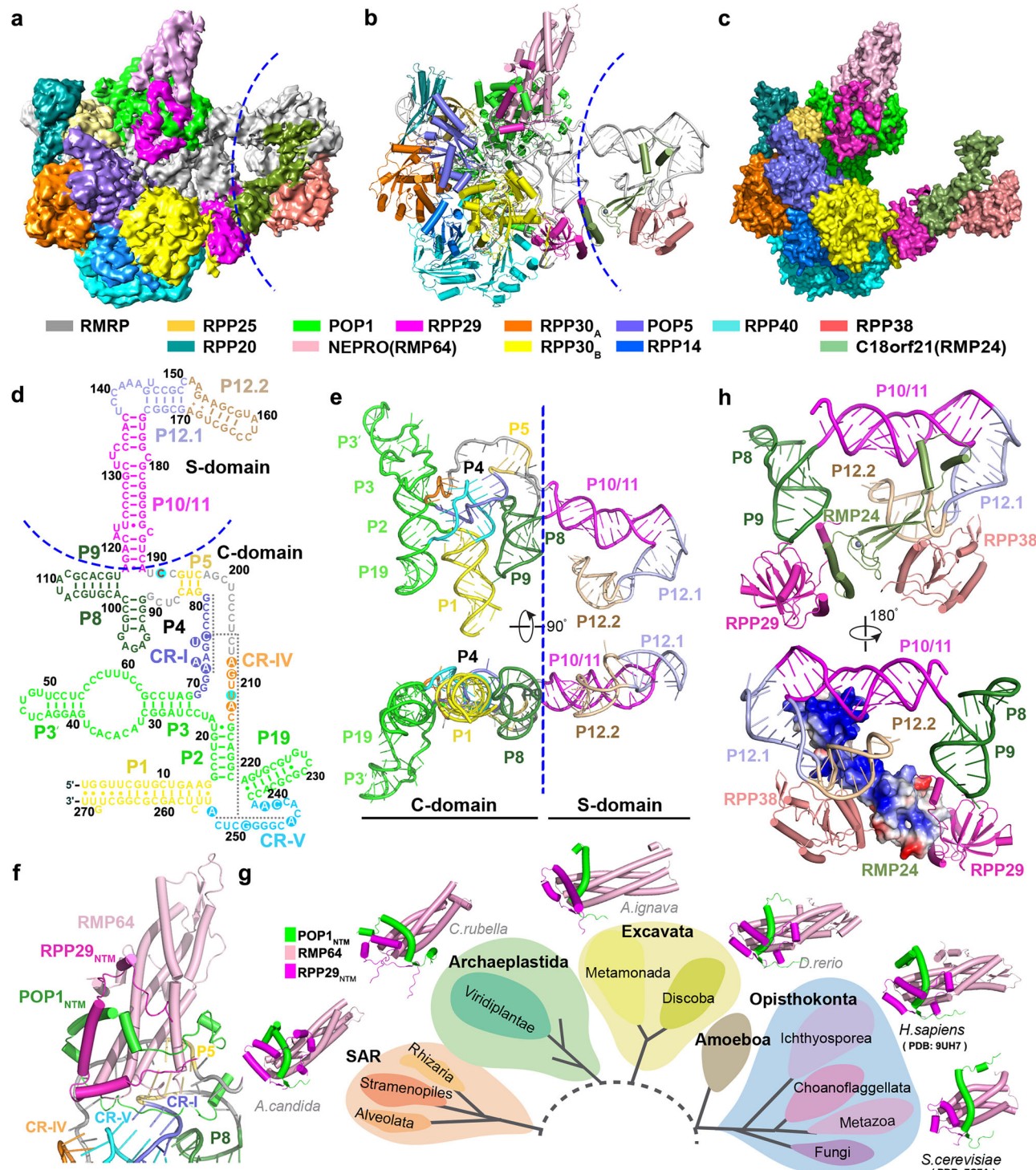

**Fig. 2 | Cryo-EM structure of human RNase MRP. a, b** The cryo-EM density map (**a**) and atomic model (**b**) of human RNase MRP. The dashed line denotes the boundary between the large lobe and the small lobe. **c** Hook-shaped protein clamp in a surface representation. The protein and RNA subunits are color-coded.
**d** Secondary structure of the RMRP RNA. Stem P1 (yellow); P2, P3, P3′, and P19 (green); P5 (yellow-orange); P8 and P9 (forest green); P10/11 (magenta); P12.1 (light blue); and P12.2 (wheat). The conserved regions CR-I, CR-IV, and CR-V are colored in slate, orange, and cyan, respectively. Universally conserved nucleotides within these regions are highlighted with shaded circles. Key −2 and +4 anchor nucleotides, U211 and C193, are highlighted with cyan circles. **e** Two orthogonal views of

the RMRP RNA structure colored as in (d). **f** The RMP64-POP1_{NTM}-RPP29_{NTM} ternary module docked onto the P5 groove of the RMRP RNA. CR-I, CR-IV, and CR-V are colored in slate, orange, and cyan, respectively. **g** Evolutionary conservation of the RMP64-POP1_{NTM}-RPP29_{NTM}-like ternary module in eukaryotic RNase MRP. AlphaFold-predicted structures from representative species are shown in cartoon representation and colored as in (**f**). **h** The large insertion in RMP24 stabilizes the small lobe of human RNase MRP. The RMP24-RPP29-RPP38 module is colored as in (**a**) and the RNA elements as in (**e**). RMP24 is shown in cartoon and electrostatic surface potential (positive: blue, negative: red) representations in the upper and lower panels, respectively.

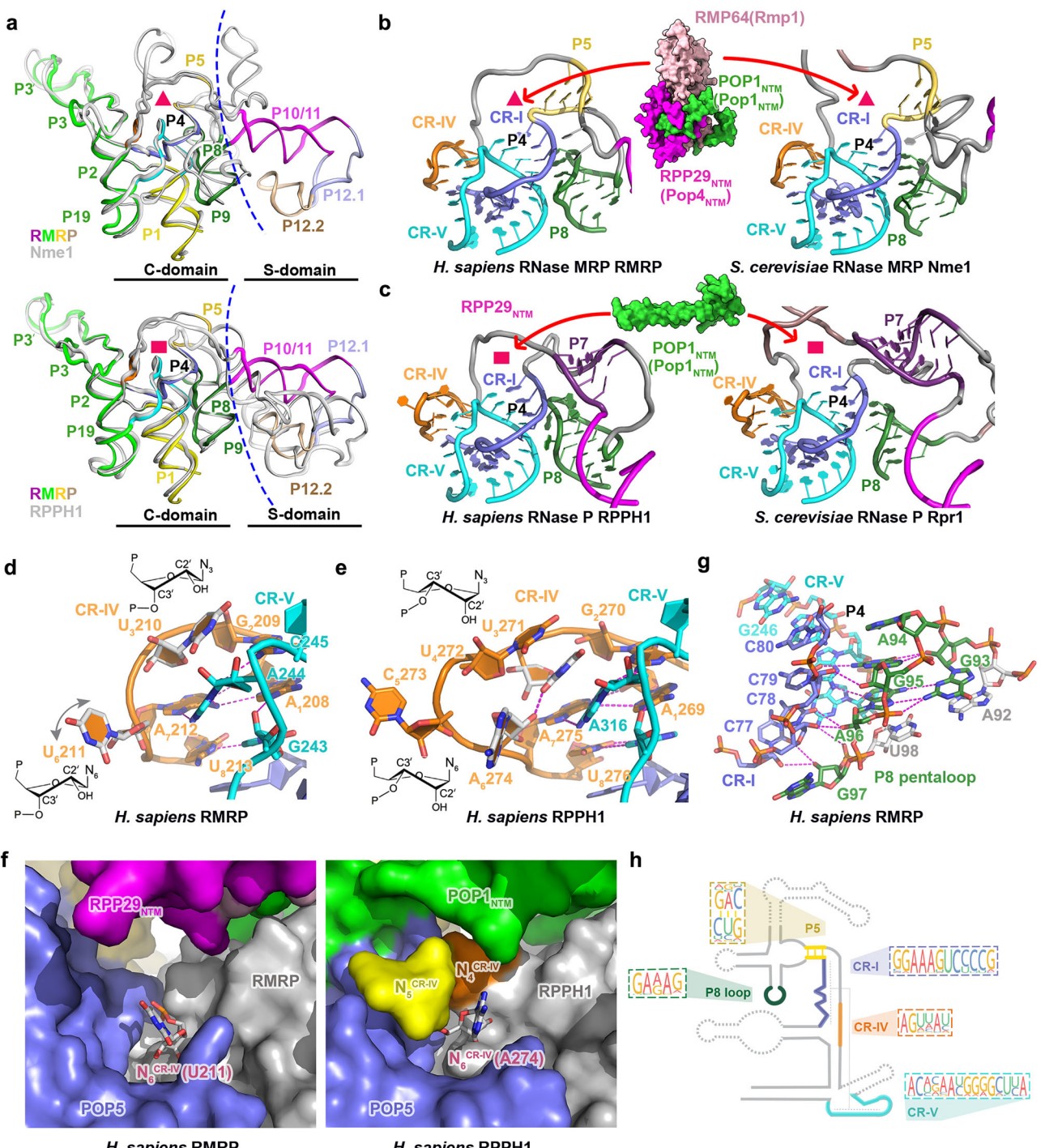

**Fig. 3 | Distinct RNA structural features in RNase P and MRP. a** Comparative Analysis of the RNA structures in RNase P and MRP. Upper panel: Comparison of human and yeast MRP RNA structures. The cradle-shaped surface formed by stems P4 and P5 that holds the RMP64–POP1$_{NTM}$–RPP29$_{NTM}$ ternary module in RNase MRP is highlighted by a red triangle. Lower panel: Comparison of human MRP RNA and human RNase P RNA. The cavity that holds the refolded POP1$_{NTM}$ in RNase P is highlighted by a red rectangle. **b** CR-IV and stems P4 and P5 form a cradle-shaped surface in RNase MRP RNA. On top of this surface sits the RMP64–POP1$_{NTM}$–RPP29$_{NTM}$ ternary module which is shown in surface representation and colored as in Fig. 2. **c** CR-IV and stems P4 and P7 form a smaller cavity in RNase P RNA. The refolded POP1$_{NTM}$ motif fits in this cavity. **d, e** The U-shaped CR-IV of human RMRP (**d**) and human RPPH1 (**e**). The sugar pucker configurations of N$_3$^{CR-IV} and N$_6$^{CR-IV} are shown. Hydrogen-bonding interactions are denoted as magenta dashed lines. **f** Comparison of N$_6$^{CR-IV} in human RMRP (left panel) and RPPH1 (right panel). N$_6$^{CR-IV} is shown in a stick representation, and the rest of the RNAs and protein components are shown in a surface representation. **g** Detailed structure of P8 pentaloop and its interactions with P4. Hydrogen bonds and electrostatic interactions are denoted as magenta dashed lines. The P8 pentaloop is colored in green, and CR-I and CR-V of P4 are colored in blue and cyan, respectively. **h** Phylogenetic conservation of RNase MRP-specific motifs. The human RMRP is shown in a simplified secondary structure with highlighted conserved elements. Logo representations of conserved element sequences across eukaryotic species from Metazoa, Fungi, Amoeba, Archaeplastida, SAR, and Excavata. The representative structures are derived from PDB 9UHA (*Homo sapiens* RNase MRP), 7C79 (*Saccharomyce cerevisiae* RNase MRP), 6AHR (*Homo sapiens* RNase P), and 6AGB (*Saccharomyce cerevisiae* RNase P)[27,48,49].

center (Fig. 3d, e, and Supplementary Fig. 14a, b). Notably, close-up structural inspection revealed that among the six CR-IV nucleotides, only one, $N_6^{CR-IV}$, adopts different conformations in human RMRP and yeast Nme1, a C2′-endo pucker in U211$^{RMRP}$ versus a C3′-endo in U266$^{Nme1}$, indicative of the dynamic nature of this nucleotide in the apo-state of RNase MRP (Fig. 3d, f and Supplementary Fig. 14a, c). In sharp contrast, due to the presence of $N_4^{CR-IV}$ and $N_5^{CR-IV}$ and the refolded POP1$_{NTM}$, $N_6^{CR-IV}$ (human A274$^{RPPH1}$, yeast A314$^{Rpr1}$) in a much more spatially constrained environment is forced to adopt a more rigid conformation with a C3′-endo pucker in the apo-state of RNase P (Fig. 3e, f, and Supplementary Fig. 14b, d). A hydrogen bond between this C2′-exo hydroxyl of $N_6^{CR-IV}$ and the base of $N_3^{CR-IV}$, whose C2′ is also in an exo configuration, further buttresses the $N_6^{CR-IV}$ conformation in RNase P (Fig. 3e and Supplementary Fig. 14b). In contrast, a C2′-endo orientation of $N_3^{CR-IV}$ in RNase MRP fails to exert this restriction on $N_6^{CR-IV}$ (Fig. 3d and Supplementary Fig. 14a). Collectively, this comparative analysis reveals a subtle nuance in CR-IV behavior in the apo-state structures, a more dynamic $N_6^{CR-IV}$ in RNase MRP versus a fixed one in RNase P.

Another salient feature uniquely conserved in RNase MRP RNAs is the pentaloop of stem P8 that folds into a compact three-layered motif with a consensus sequence 5′-GARAR-3′ (R: purine) (Fig. 3g and Supplementary Fig. 14e). In human RMRP pentaloop ($_{93}$GAGAG$_{97}$), G93 and A96 form a Hoogsteen base pair as the foundation layer, on which sequentially stack G95 and G94 (Fig. 3g). This purine-rich pentaloop makes close contact with the minor groove of the P4 stem in such an orientation that nucleotides G95 and A96 from the P8 pentaloop mediate an intricate hydrogen-bonding network with stem P4, 'gluing' the pentaloop to the catalytic pseudoknot (Fig. 3g). Consequently, these extensive interactions fix G97, the last nucleotide in the pentaloop, in a flipped-out configuration to pack against the P4 backbone (Fig. 3g). A similar mode of interactions between the P8 pentaloop and stem P4 is conserved in yeast Nme1 (Supplementary Fig. 14e). On the contrary, the non-purine-rich P8 loops in human RPPH1 and yeast Rpr1 RNAs make none or limited contacts with the P4 minor groove, likely rendering a less stable connection between the P8 loop and the pseudoknot core (Supplementary Fig. 14f, g).

To explore the evolution of these unique RNA elements, we performed phylogenetic and secondary structure analyses on RNase MRP RNAs from diverse eukaryotic phyla (Supplementary Figs. 15 and 16). These analyses revealed that the unique RNA features found in human and yeast RNase MRP structures—a three-base-pair P5 stem immediately adjacent to CR-I, a P8 pentaloop with a 5′-GARAR-3′ consensus sequence, and a six-nucleotide CR-IV—are all highly conserved in RNase MRP from protists to metazoans (Fig. 3h and Supplementary Figs. 15 and 16). On the contrary, unique RNA features found in human and yeast RNase P—the replacement of P5 with a P7 stem at least several nucleotides downstream of CR-I and an eight-nucleotide CR-IV—are evolutionarily conserved only in RNase P (Supplementary Fig. 17a, b). Notwithstanding these marked variations, a common theme is that these evolutionarily conserved RNA motifs, either unique to eukaryotic RNase P or to RNase MRP, all gather in the close vicinity of the catalytic pseudoknot, supporting a scenario that these unique RNA structural features had evolved under different purifying selection pressures for processing distinct RNA substrates. In accordance with this idea, mutations, insertions, or deletions in stem P5, CR-IV, and the P8 pentaloop of human RMRP have been found in patients with phenotypes of CHH and AD, underscoring the physiological significance of these unique RNA elements in RNase MRP (Supplementary Fig. 18a, b)[9–12,14,15].

## A double-anchor mechanism for ssRNA substrate recognition by RNase MRP

To elucidate the substrate recognition mechanism of RNase MRP, we performed in vitro cleavage assays using the human ITS1-2′ substrate

(Fig. 4a). The results revealed a major cleavage event mapped to a 5′-*CGUU-3′ motif (*: cleavage site) (Fig. 4a). Then we fitted the human ITS1-2′ substrate into our RNase MRP structure, using the yeast RNase MRP-substrate complex structure as a guide[27]. Despite notable differences between human and yeast RNase MRP structures, the overall architecture of the substrate-binding groove is remarkably conserved, and the ITS1-2′ substrate can be snugly fit into the human RNase MRP substrate-binding site by and large with the same set of interactions as the yeast ITS1 substrate, suggestive of a common substrate recognition mechanism (Fig. 4b and Supplementary Fig. 19). In both human and yeast RNase MRP structures, the RMP64–POP1$_{NTM}$–RPP29$_{NTM}$ (Rmp1–Pop1$_{NTM}$–Pop4$_{NTM}$ in the yeast complex) ternary module, stabilized by the P5 stem and its adjacent polynucleotides from P10 to CR-IV as well as the P8 pentaloop, creates a deep groove atop the catalytic pseudoknot suitable for accommodating an extended, ssRNA substrate (Fig. 4b and Supplementary Fig. 19a). Notably, via electrostatic and hydrophobic contacts RPP29$_{NTM}$ and RMP64 residues on the inner surface of this deep groove help shape the ssRNA substrate in a conformation structurally mimicking the to-be-processed strand of the pre-tRNA acceptor stem in the RNase P active site (Supplementary Fig. 19b, c)[27,48,49]. Consistent with this structural mimicry, mutations in the RPP29$_{NTM}$ (R41A, Q50A, and R53A) disrupted RNase MRP-mediated pre-rRNA processing in cells (Supplementary Fig. 20a–c).

A further close-up examination of this substrate-binding groove unveils that it functions as a measuring device that could secure a six-nucleotide ssRNA substrate between two unique anchors (Fig. 4b and Supplementary Fig. 19a). ~36 Å apart, these anchors mediate specific interactions with substrate nucleotides at positions −2 and +4 relative to the cleavage site (Fig. 4b and Supplementary Fig. 19a). The six-nucleotide, U-shaped CR-IV is the −2-anchor, with $N_6^{CR-IV}$ (human U211$^{RMRP}$, yeast U266$^{Nme1}$) stacking with the substrate −2-nucleotide (Fig. 4b and Supplementary Fig. 19a, d). This base stacking is continuously extended until it reaches the +4-nucleotide, which flips away from the rest of the substrate and plugs its base into a deep pocket, the +4-anchor, at the other side of the substrate-binding groove (Fig. 4b and Supplementary Fig. 19a). The imidazole ring of a histidine residue in POP1$_{NTM}$ (human His146$^{POP1}$ and yeast His103$^{Pop1}$) serves as the floor of the pocket, stably holding the base of the +4-nucleotide (Fig. 4c and Supplementary Fig. 19e). In human RNase MRP, this histidine residue is firmly stapled in an optimal position for +4-nucleotide packing through an array of immediately adjacent hydrogen-bonding interactions, including two with the flipped-out G97 in the P8 pentaloop (Fig. 4c). An alanine substitution of His146$^{POP1}$ resulted in a substantial increase in pre-cleaved ITS1 signals, underscoring the importance of this residue in pre-rRNA processing (Fig. 4d, e and Supplementary Fig. 20d, e). At the bottom of the pocket, a pyrimidine (human C193$^{RMRP}$, yeast U226$^{Nme1}$), one nucleotide downstream of the P5 stem, rotates away from the stem and fits into a cavity formed by RMP64 and POP1$_{NTM}$ with its base lined up by two hydrogen bonds with RMP64 Tyr70 and Asn74 for a head-to-head recognition for the substrate +4 nucleotide (Fig. 4c and Supplementary Fig. 19e). The dimension of this C193$^{RMRP}$ cavity could not accommodate a purine nucleotide, consistent with the fact that both human RMRP and yeast Nme1 possess a pyrimidine in this position (Fig. 4c and Supplementary Fig. 19e).

In line with structural insights, RNase MRP activity assays using the human ITS1 substrate demonstrated a preference for a pyrimidine at the +4 position, consistent with the spatial constraints of the +4 cavity (Fig. 4a). In yeast, POP4 Arg28 mediates substrate recognition through a "pseudo–Watson–Crick" interaction, complemented by a hydrogen bond with the sugar-ring hydroxyl group of C2 (Supplementary Fig. 19b). Similarly, in humans, the conserved RPP29 Arg53 residue across vertebrates is predicted to interact with the +3 pyrimidine nucleotide of the substrate, forming a pseudo base pairing structure (Supplementary Figs. 11a, 19b, c). Consistent with this observation, overexpression of the RPP29$^{R53A}$ mutant led to increased

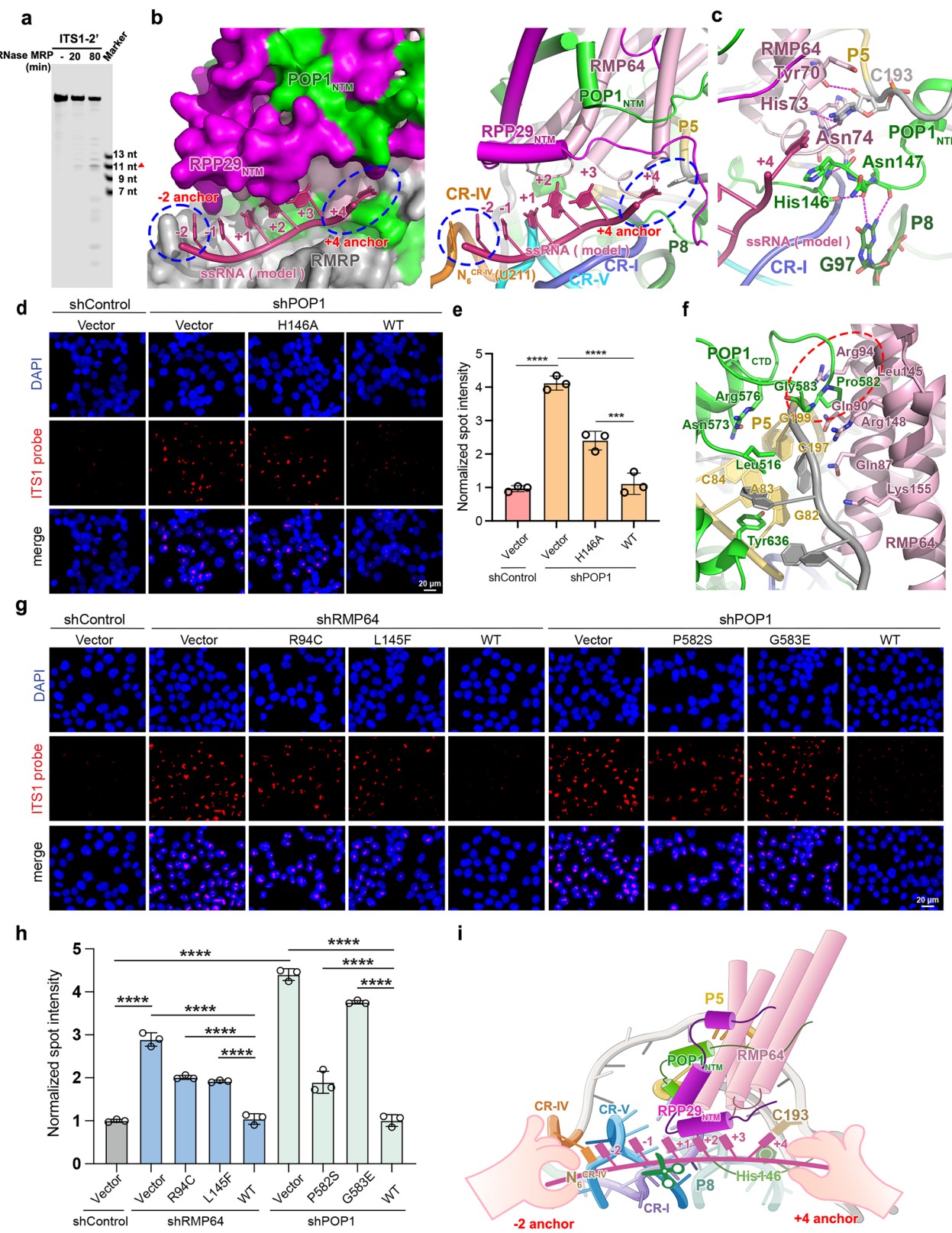

accumulation of pre-rRNA ITS1 compared to wild-type RPP29 (Supplementary Fig. 20a–c). Taken together, these findings demonstrate that RPP29 Arg53 contributes to substrate specificity by stabilizing the +3 pyrimidine, facilitating the flipped-out conformation of the +4 pyrimidine into the +4-anchor pocket.

In the vicinity of the +4-anchor, a short helix from POP1$_{CTD}$ packs on stem P5 from the opposite side of the RMP64-P5 interface, and the

extended loop C-terminal to this POP1 helix fits into a depression on RMP64 to tightly lock the P5 stem between RMP64 and POP1$_{CTD}$ (Fig. 4f). It is noteworthy that mutations of RMP64 and POP1 at this interface that presumably would deform the spatial arrangement of the +4-anchor have been found in patients with AD phenotypes[45,46,53,54]. Consistent with the clinical observations, expression of these mutations failed to rescue the pre-rRNA processing defect in RMP64 or

**Fig. 4 | A 'double-anchor' substrate-recognition mechanism of RNase MRP. a** In vitro cleavage assay of human ITS1-2' substrate (5'-FAM-GAGUCCGGUCCCGU UUGCUGUCUCGUCU-3'). One major cleavage site is denoted by a red triangle. The experiments were analyzed using at least three biological replicates. **b** The substrate-binding groove of human RNase MRP. Surface (left) and cartoon (right) representations of the substrate-binding groove with a modeled ssRNA substrate. The substrate is shown in a cartoon representation and colored in warm pink. The −2 and +4 anchors are denoted by blue dashed circles. **c** Close-up view of the +4 anchor of human RNase MRP. **d, e** Immunofluorescence analysis of pre-rRNA processing defects caused by the alanine substitution of His146$^{POP1}$ (**d**) with the quantification shown on the right (**e**). Scale bar, 20 μm. At least 100 cells were analyzed.

**f** The cartoon representation of disease-related residues in POP1 and RMP64 near the substrate-binding site. The disease-related residues are highlighted with a red dashed circle. Proteins and the RMRP RNA are colored as in Fig. 2. **g, h** Immuno-fluorescence analysis of pre-rRNA processing defects caused by disease-related mutations in RMP64 or POP1 (**g**) with the quantification shown in the lower panel (**h**). Scale bar, 20 μm. At least 100 cells were analyzed. **i** A schematic cartoon of the 'double-anchor' substrate-recognition mechanism by RNase MRP. The −2 and +4 anchors cooperate to anchor the ssRNA substrate into the catalytic site. For all experiments, data were shown as mean ± SD from three independent replicates. Significance was determined using One-way ANOVA. ***$p < 0.001$, ****$p < 0.0001$.

POP1 knockdown cells, underscoring the essential role of the +4-anchor in pre-rRNA processing by RNase MRP (Fig. 4g, h and Supplementary Fig. 20f).

Multiple sequence alignment revealed that the −2-anchor, CR-IV, is rigorously six-nucleotide long in all RNase MRP RNAs and that key +4-anchor defining residues, a histidine, tyrosine, or phenylalanine in POP1$_{NTM}$ and a pyrimidine in the RNA, are conserved across diverse eukaryotic phyla from protists to metazoans (Fig. 3h, Supplementary Figs. 11a, b and 15). Furthermore, unique RNase MRP-specific features, unveiled in both human and yeast RNase MRP structures, that stabilize these anchors—stem P5, the P8 pentaloop, and the RMP64-POP1$_{NTM}$-RPP29$_{NTM}$ ternary module—have also remained highly preserved across eukaryotic lineages during evolution, highlighting the concerted evolution of an integrated ssRNA substrate-binding mechanism (Fig. 4i). Given that all key anchor-defining residues—U211$^{RMRP}$/U266$^{Nme1}$, His146$^{POP1}$/His103$^{Pop1}$, and C193$^{RMRP}$/U226$^{Nme1}$—adopt almost identical spatial arrangement in both human and yeast RNase MRP structures, our evolutionary analysis suggests that all eukaryotic RNase MRP ribozymes very likely employ a common, evolutionarily conserved 'double-anchor mechanism' for ssRNA substrate recognition.

## Distinct substrate recognition mechanisms of RNase P and MRP —an 'anchor−substrate matching' hypothesis

There exists a puzzle about the CR-IV region. In all available structures of RNase P and MRP in complex with pre-cleaved substrates, the −2-anchor nucleotide N$_6$$^{CR-IV}$ adopts the same conformation to stack with the substrate −2 nucleotide (Supplementary Fig. 21a)[24,27,48]. This brought up an intriguing question—why the CR-IV regions of RNase P and MRP have stringently preserved eight and six nucleotides during evolution—given that they employ the same mechanism to anchor the substrate −2 nucleotide (Supplementary Figs. 15 and 17 and 21a−d)? A clue to solve this puzzle stemmed from our observation of −2-anchor nucleotide N$_6$$^{CR-IV}$'s distinct behaviors in the apo-state ribozymes—a dynamic conformation in RNase MRP versus a fixed one in RNase P (Fig. 3d, e and Supplementary Fig. 14a, b). This inspired us to come up with an 'anchor-substrate matching' hypothesis for the substrate recognition by RNase P and MRP, in which we proposed that a dynamic −2-anchor nucleotide N$_6$$^{CR-IV}$ is required to accommodate the flexible ssRNA substrate in RNase MRP, while a prefixed N$_6$$^{CR-IV}$ is preferred for securing the rigid pre-tRNA in RNase P. We swapped the CR-IV regions between human RMRP and RPPH1 RNAs and overexpressed these mutant RNAs in 293 T cells to test this idea (Fig. 5a and Supplementary Fig. 22a, b). Markedly aligned with the hypothesis, FISH and quantitative RT-PCR analyses revealed a substantial accumulation of pre-cleaved ITS1 or pre-tRNA signals in cells expressing mutant RMRP and RPPH1 with the swapped CR-IV (Fig. 5b−d). These results suggest that the eight- and six-nucleotide CR-IV with distinct dynamic behaviors in N$_6$$^{CR-IV}$ likely have been stringently selected during evolution by RNase P and MRP, respectively, to process different substrates—rigid pre-tRNA versus flexible ssRNA.

What is the mechanistic underpinning of this 'anchor-substrate matching' hypothesis? To address this question, we closely

reexamined all the available RNase P and MRP structures in the apo and substrate-bound states (Supplementary Fig. 21a)[23,24,27,30,48,49,55]. In RNase P, CR-IV and POP1$_{NTM}$ together constitute a pre-formed anchor that can seamlessly accommodate the rigid single-double strand junction region of the pre-tRNA acceptor stem with the overhang −1 and −2-nucleotides capped by the rigid N$_6$$^{CR-IV}$ (Fig. 5e and Supplementary Fig. 19c)[24,48]. About 51 Å away on the other side of the RNase P complex, the TψC and D loops of pre-tRNA fit into another pre-organized anchor formed by the CR-II/III T-loops in the S-domain of the RNA (Fig. 5e and Supplementary Fig. 12c, d)[24,48]. In aggregate, the recognition of a rigid pre-tRNA substrate by these two pre-formed anchors of RNase P is reminiscent of a ski boot fitting into a ski in a precise manner so that the scissile phosphate of the pre-tRNA substrate resides right on the catalytic center optimal for cleavage.

Structural superposition unveiled identical configurations of the catalytic center in both RNase P and MRP complexes, suggesting the same RNA-based $S_N2$-type catalytic mechanism for both ribozymes (Fig. 5e and Supplementary Fig. 21b-d). This implies that even though RNase MRP processes ssRNAs, these substrates should adopt the same configuration as the to-be-processed strand of the pre-tRNA acceptor stem in the active site of RNase P. Indeed, the −2 to +3 nucleotides of the ITS1 substrate in the RNase MRP active site highly resemble their pre-tRNA counterparts in the RNase P active center (Supplementary Fig. 21a). The RNase MRP unique 'double-anchor mechanism' secures the backbone of −2 to +3 substrate nucleotides in the catalytic center in the same manner as the to-be-processed strand of the pre-tRNA acceptor stem; on one end of the substrate-binding groove N$_6$$^{CR-IV}$ caps the continuous-stacked −2 to +3 substrate nucleotides, while on the other end a deep pocket formed by both RNA and the RMP64-POP1$_{NTM}$-RPP29$_{NTM}$ ternary module grabs and fixes +4-nucleotide (Fig. 5e and Supplementary Fig. 19b, c). Furthermore, residues from the RMP64-POP1$_{NTM}$-RPP29$_{NTM}$ module function like the complementary strand of the acceptor stem in pre-tRNA, forming pseudo-base-pair contacts with substrate nucleotides on the +1 to +3 positions, structurally resembling the double-stranded conformation of the pre-tRNA acceptor stem (Fig. 5e and Supplementary Fig. 19b, c). Together, this structural mimicry arranges the scissile phosphate of the ssRNA substrate right on the active site of RNase MRP for catalysis (Fig. 5e and Supplementary Fig. 21a).

It should be noted that a big difference—rigidity—exists between the substrates of RNase P and MRP. Before being recognized by RNase P, pre-tRNA is already a pre-folded, rigid RNA with a well-defined 3D structure. But, the ssRNA substrate, in contrast, should adopt a much more flexible conformation before entering the substrate-binding groove of RNase MRP. This rigidity difference in substrates provides a plausible explanation for the 'anchor-substrate matching' hypothesis. We propose that an eight-nucleotide CR-IV with a rigid N$_6$$^{CR-IV}$ would be thermodynamically favorable to match the shape of a pre-formed pre-tRNA acceptor stem (Fig. 5e). By contrast, a six-nucleotide CR-IV with a flexible N$_6$$^{CR-IV}$ is likely essential for the initial contact of the substrate-binding groove of RNase MRP with a less ordered ssRNA substrate that is subsequently induced into a continuously packed configuration

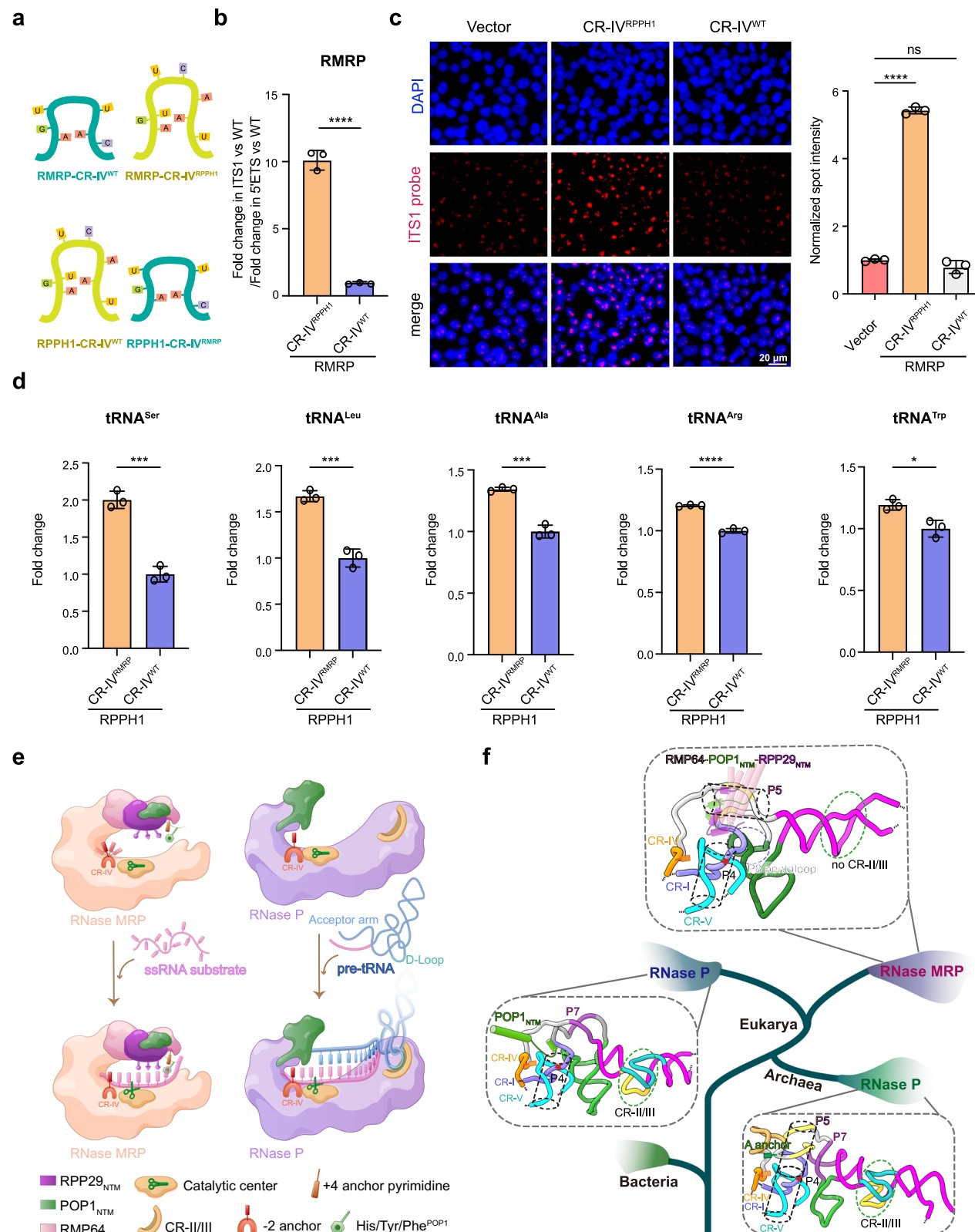

(Fig. 5e). We speculate that, to process the ssRNA substrate by the same RNA-based catalytic mechanism inherited from prokaryotic RNase P ancestors, RNase MRP had evolved the aforementioned unique RNA motifs and protein factors to module the ssRNA substrate into a conformation mimicking the to-be-processed strand of the pre-tRNA acceptor stem (Fig. 5e, Supplementary Figs. 19b, c and 21d)[27]. This evolutionarily conserved structural mimicry strategy has

remained invariant in eukaryotes since its emergence in primitive species, highlighting its fundamental significance in substrate recognition by eukaryotic RNase MRP (Fig. 5e and Supplementary Fig. 21a).

## Discussion

The cryo-EM structure of human RNase MRP, integrated with comparative and evolutionary analyses, provides novel insights into the

**Fig. 5 | Distinct substrate-recognition mechanisms of RNase P and MRP.**
**a** Schematic diagram of the domain-swap strategy of CR-IV. Chimeric RNAs were generated by exchanging the CR-IV sequences between human RMRP and RPPH1. **b**–**d** HEK293T cells were infected with an empty vector or CR-IV-swapped mutants of RMRP and RPPH1. Quantitative RT-PCR (**b**) and FISH (**c**) analyses reveal a substantial accumulation of pre-cleaved ITS1 or pre-tRNA signals (**d**) in cells expressing the mutant RMRP and RPPH1 with swapped CR-IV. Scale bar, 20 μm. At least 100 cells were analyzed. For all experiments, data were shown as mean ± SD from three independent replicates. Significance was determined with a two-tailed Student's t-test. *p < 0.05, ***p < 0.001, ****p < 0.0001. **e** Schematic illustration of the evolutionarily conserved structural mimicry strategy employed by RNase MRP-mediated ssRNA processing, in comparison with RNase P-mediated pre-tRNA processing. RNase MRP utilizes a novel 'double-anchor' mechanism to stably engage flexible ssRNA substrates, whereas RNase P recognizes the pre-tRNA substrate in a more rigid manner. **f** Proposed evolutionary model of eukaryotic RNase P and MRP. Illustrative cladogram and representative RNA 3D structures showing the key modifications of both RNA and protein components along the evolutionary history, in which eukaryotic RNase P and MRP originated from an ancestral ribozyme resembling archaeal RNase P. Red star denotes the active site. Cylinders in black dashed lines indicate the relative positions of stems P4 and P5. Representative RNA 3D structures are based on PDB 6K0A (*Methanococcus jannaschii* RNase P), 6AHR (*Homo sapiens* RNase P), and 9UHA (*Homo sapiens* RNase MRP).

architecture, substrate recognition, and evolution of this essential eukaryotic ribonucleoprotein complex. Our study reveals that eukaryotic RNase MRP RNA, a duplicate of the ancestral RNase P RNA, had evolved novel, unique structural features, including a repositioned stem P5, a more dynamic six-nucleotide CR-IV, and a purine-rich P8 pentaloop, all around the catalytic pseudoknot core that has been invariant since the RNA world[3,16,20,32]. These unique RNA elements, in concert with the emergence of a novel RMP64–POP1$_{NTM}$–RPP29$_{NTM}$ module, create a dedicated substrate-binding groove that secures flexible ssRNA substrates through a novel 'double-anchor' mechanism. In this configuration, the ssRNA substrate is stabilized by conserved interactions at the −2 and +4 nucleotide positions, allowing precise endo-nucleolytic cleavage by the catalytic core (Fig. 5e). This strategy contrasts with that of RNase P, which employs an eight-nucleotide CR-IV and the refolded POP1$_{NTM}$ to recognize the rigid acceptor stem of pre-tRNAs (Fig. 5e).

A comparative analysis of archaeal RNase P and eukaryotic RNase P and MRP supports an evolutionary model in which eukaryotic RNase P and MRP originated from an ancestral ribozyme resembling archaeal RNase P, subsequently diverging into two distinct complexes through the coordinated remodeling of both RNA and protein components (Fig. 5f and Supplementary Fig. 23a, b). This ancestral ribozyme contains five protein factors, but it employs two pre-organized RNA-based anchors to recognize rigid pre-tRNA substrates—an eight-nucleotide CR-IV and an invariant adenine located between the P5 and P15 stems in the C-domain form a pre-formed anchor that recognizes the pre-tRNA acceptor stem, while the conserved CR-II/III T-loop anchor in the S-domain recognizes the TψC and D loops of pre-tRNA substrates (Fig. 5f and Supplementary Fig. 23a, b). It is noteworthy that, in this configuration, an essential short stem—P5—aligns coaxially with P4, playing a key role in accommodating the to-be-processed strand of the pre-tRNA acceptor stem into the junction groove between P4 and P5, right above the catalytic pseudoknot optimal for cleavage (Fig. 5f and Supplementary Fig. 23a, b). Alongside eukaryote evolution, we propose that the necessity to process single-stranded pre-rRNA substrates in addition to pre-tRNAs by the same RNA-based $S_N2$ catalytic mechanism drove the ancestral ribozyme complex to acquire novel RNA and protein components, branching into two distinct evolutionary trajectories. In one branch, eukaryotic RNase P maintained most of the ancestor RNA motifs, including the eight-nucleotide CR-IV and the CR-II/III T-loop anchor, but it dismantled the P5 stem and instead employed a newly acquired protein, POP1, together with CR-IV to generate a novel RNA-protein-aided anchor to recognize the rigid acceptor stem of pre-tRNA (Fig. 5f and Supplementary Fig. 23a, b)[6,52]. By contrast, along the other trajectory, the duplicated RNA in eukaryotic RNase MRP underwent substantial modifications compared to its ancestor—removal of the CR-II/III T-loops, length reduction of CR-IV from eight to six nucleotides, and the repositioning of stem P5 in a flipped-out conformation away from the original coaxially stacked P4-P5 orientation in the ancestral RNA (Fig. 5f and Supplementary Fig. 23a, b). This marked RNA reconfiguration was accompanied by the emergence of RNase MRP-specific protein subunits RMP64 and RMP24 and

the refolding of POP1 and RPP29 to evolve a novel substrate-binding platform suitable for accommodating flexible ssRNA substrates in the catalytic center via a structural mimicry strategy (Fig. 5f and Supplementary Fig. 19b, c). We speculate that this RNase P-to-MRP adaptation was likely driven by the purifying selection pressure of accommodating biophysically distinct substrates rather than by random structural drift during evolution[6,16,56,57].

The structural flexibility of RNase MRP to accommodate short ssRNA substrates is reflected in its broader substrate range. Unlike RNase P, which exhibits stringent specificity for rigid pre-tRNA substrates, RNase MRP is capable of processing a diverse array of ssRNAs, consistent with previous reports and demonstrated by our in vitro cleavage assays (Supplementary Fig. 24a, b)[1,4,26,28]. This substrate promiscuity implies a broader role of RNase MRP in eukaryotic RNA metabolism, potentially influencing RNA maturation, surveillance, and decay pathways in response to developmental or environmental cues[28,58–63]. The clinical relevance of this substrate recognition mechanism is underscored by the fact that disease-causing mutations in *RMRP*, *RMP64*, and *POP1* frequently map to residues critical for RNA anchoring or structural stabilization (Fig. 4f–h and Supplementary Fig. 18a, b)[9,10,12,45,46,54,64]. These mutations are associated with skeletal developmental disorders such as CHH and AD, highlighting the biological relevance of the substrate recognition platform we have defined.

In summary, this study establishes a unified structural framework for understanding how RNase MRP evolved from an ancestral ribozyme that processes rigid pre-tRNAs into one to process flexible ssRNA substrates. This transition involves the retention of a conserved catalytic RNA core and the acquisition and functionalization of novel RNA and protein elements. These findings lay the foundation for future investigations into the regulation of RNase MRP activity, the breadth of its RNA targets across different tissues and cell states, and the structural principles that govern its substrate specificity and functional integration into eukaryotic RNA metabolism.

## Methods
### Identification of the unique subunits of eukaryotic RNase MRP with a structural-based search pipeline
Human RNase MRP shares most of its protein subunits with human RNase P, including POP1, POP5, RPP14, RPP20, RPP25, RPP29, RPP30, RPP38, and RPP40[65–69]. However, it remains unclear whether human RNase MRP contains homologs of *S. cerevisiae* Rmp1 and Snm1 that are unique to yeast RNase MRP[6,33]. To identify potential Rmp1 and Snm1 homologs in human and other eukaryotic genomes, we initially used the protein sequences of Rmp1 and Snm1 as queries in BLAST searches against the NCBI and UniProt protein databases; however, no promising candidates were identified. Consequently, we implemented a structure-based search pipeline to extend our search in human and other eukaryotic genomes.

First, typical structural motifs in Rmp1 and Snm1 were isolated and used as query structures on the DALI server (https://ekhidna2.biocenter.helsinki.fi/dali/) to search against the AlphaFold

database[70,71]. Species included in this search were *H. sapiens*, *M. musculus*, *X. laevis*, *D. rerio*, *D. melanogaster*, *D. discoideum*, and *A. thaliana*. Proteins that exhibited significant structural similarity (DALI Z-score > 6) were retained as homologous candidates (Supplementary Data 1 and 2). Next, candidate structures were manually compared to those of Snm1 and Rmp1. For human homolog candidates, we conducted a detailed one-by-one comparison—examining the zinc finger domain (for Snm1 homologs) or the four-helix bundle domain (for Rmp1 homologs)—to confirm the most relevant matches.

Since the DALI search was limited to a subset of eukaryotic species, we further employed the Foldseek method (https://foldseek.berkeley.edu/) to expand our search to encompass all available eukaryotic species. Both the yeast proteins and the human candidates were used as queries to identify additional homologs based on structural similarity (Supplementary Data 3). Manual structural comparisons were then performed on the protein candidates—applying criteria of a prob value > 0.5, a score value > 0.5, and a score value < 1 × 10$^{-3}$—to ensure the presence of the zinc finger domain in Snm1 homolog candidates and the four-helix bundle structure in Rmp1 homolog candidates across multiple eukaryotic species. Sequence alignments of the identified homologous proteins were generated using MAFFT[72] and subsequently adjusted manually based on their structure-based alignments.

## Construction of stable gene knockdown cell lines

To generate stable gene knockdown cell lines, short hairpin RNAs (shRNAs) targeting human RMP64, RMP24, or POP1 were introduced into HEK293T cells. The shRNA oligos were designed through an online tool of siRNA target finder (Ambion) and synthesized by Sangon Biotech, as listed in Supplementary Data 5. The constructs were cloned into a modified pGreen-puro lentiviral vector containing both an enhanced green fluorescent protein (EGFP) reporter and a puromycin resistance gene. A non-targeting shRNA served as the negative control. For lentiviral production, HEK293T cells were co-transfected with the pGreen-puro-shRNA plasmid and packaging plasmids (pVSVG and pDR8.9) using the X-tremeGENE HP transfection reagent (Roche). Virus-containing supernatant was collected 48 h post-transfection, filtered through a 0.45-μm syringe filter, and used to infect target cells.

HEK293T cells were seeded in 6-well plates and transduced with the lentiviral particles. After 48 h, the cells were selected with 2 μg mL$^{-1}$ puromycin for 2 days to ensure stable integration of the shRNA construct. Knockdown efficiency was subsequently assessed by quantitative PCR (qPCR), and only cells demonstrating sufficient knockdown were used in downstream experiments.

For the rescue experiments, both wild-type and mutant forms of RMP64 and POP1 were cloned into a modified pminiCMV-mCherry expression vector. Mutant constructs were generated via site-directed mutagenesis to introduce specific mutations in RMP64 (R94C and L145F) or POP1 (P582S, G583E, and H146A). For rescue transfection, wild-type, mutant, and control expression vectors were co-transfected with pVSVG and pDR8.9 to produce lentiviral particles. Cells stably expressing shRNAs against RMP64 or POP1 were seeded in 6-well plates and transduced with these lentiviruses. After 24 h, the medium was replaced with fresh media, and the cells were incubated for an additional 36 h. Transfection efficiency was evaluated by mCherry fluorescence, and re-expression of the wild-type or mutant genes was confirmed by qPCR. All rescue experiments were performed in triplicate, with data normalized to their respective control groups.

## Gene expression analysis

Total RNA was isolated from $1 \times 10^6$ cells using TRIzol reagent (Invitrogen). Complementary DNA (cDNA) was synthesized with the HiScript III reverse transcriptase reagent (Vazyme) according to the manufacturer's instruction, and the resulting cDNA served as the template for qPCR with gene-specific primers (synthesized by Sangon Biotech), as listed in Supplementary Data 5.

qPCR was performed on a QuantStudio™ 6 Flex system (Thermo Fisher, 4485697). Each 20 μL reaction comprised 10 μL of ChamQ Universal SYBR qPCR Master Mix (Vazyme, Q711-02), 1 μL of cDNA, and 0.5 μM each of the forward and reverse primers targeting the genes of interest and the housekeeping gene (*18S* or *GAPDH* for HEK293T cells and *Hprt1* for mBMSCs). Thermal cycling was carried out with an initial denaturation at 95 °C for 10 minutes, followed by 40 cycles of 95 °C for 15 seconds and 60 °C for 1 minute. Relative gene expression levels were calculated using the $2^{-\Delta\Delta C_T}$ method with normalization to the housekeeping gene. All reactions were performed in triplicate. Primer sequences are provided in Supplementary Data 5 and the primers were synthesized by Sangon Biotech.

## RNA FISH analysis

RNA fluorescence in situ hybridization (RNA FISH) was performed using the RNA FISH kit (Beyotime, R0306S) following the manufacturer's instruction. HEK293T cells were seeded in 48-well plates and cultured until reaching 70-80% confluence. Cells were fixed on ice with 4% paraformaldehyde in 1×PBS for 30 minutes, then washed twice with 1 × PBS. Permeabilization was performed using 0.25% Triton X-100 in 1×PBS for 15 min. Hybridization was carried out overnight at 37 °C using Cy5-labeled ITS1 probes (5′-TACGAGGTCGATTTGGC-GAGGGCGCT-3′). The probe sequence was designed as described previously[4]. Post-hybridization, cells were washed twice with 2 × Saline-Sodium Citrate (SSC) buffer at 37 °C, followed by two washes with 1 × SSC buffer at room temperature, and then stained with DAPI (0.1 μg mL$^{-1}$). Fluorescence images were captured using an Operetta® High Content Imaging System (PerkinElmer), and fluorescence intensity was quantified using Harmony software.

## Polysome profiling assay

HEK293T Cells were cultured in DMEM until 80-90% confluence. Before harvesting, cells were treated with 100 μg mL$^{-1}$ cycloheximide (CHX) (MCE, HY-12320) for 20 min. ~$4 \times 10^6$ cells were collected, washed with cold PBS, and lysed in a buffer containing 5 mM Tris-HCl (pH 7.4), 1.5 mM NaCl, 2.5 mM MgCl$_2$, 1% Triton X-100, and 100 μg mL$^{-1}$ CHX to inhibit translation elongation. The lysate was clarified by centrifugation at 14,000 × g for 10 min at 4 °C to remove the cellular debris. Protein equivalent to 1 mg was layered onto a 5–50% (w/v) sucrose gradient prepared in a polysome gradient buffer (20 mM Hepes-KOH, pH 7.4, 150 mM NaCl, 5 mM MgCl$_2$, and 10 μg mL$^{-1}$ CHX). The gradients were ultracentrifuged at 222,000 × g for 3 h at 4 °C using a Beckman SW41 Ti rotor. Following centrifugation, the gradients were fractionated into 400 μL aliquots using a Piston Gradient Fractionator (Biocomp), and ribosomal profiles were monitored by measuring absorbance at 254 nm.

## Nascent protein synthesis assay

Nascent protein synthesis in HEK293T cells was evaluated using the BeyoClick™ HPG Protein Synthesis Kit with Alexa Fluor 555 (Beyotime, P1206S), which employs click chemistry to specifically detect newly synthesized proteins. HEK293T cells were seeded in 96-well plates at a density of $1 \times 10^4$ cells per well and incubated overnight in complete DMEM supplemented with 10% fetal bovine serum. Following overnight culture, cells were incubated in a methionine-free medium (Beyotime, C0891) for 1 h before the addition of 100 μM L-homopropargylglycine (HPG), a methionine analog that is incorporated into nascent proteins, for 1 h at 37 °C. All subsequent steps were performed at room temperature. Cells were then fixed in 4% paraformaldehyde in PBS for 15 min, followed by permeabilization with 0.25% Triton X-100 in PBS for 10 min. The click reaction was performed by incubating the cells with a reaction mixture containing Alexa Fluor 555 azide and 4 mM CuSO$_4$, as per the manufacturer's instruction, for

30 min. After washing three times with PBS, cells were stained with DAPI for 10 minutes and imaged using a high-content microscope (PerkinElmer). Fluorescence intensity was quantified using the Harmony system.

### Generation of Rmp64-knockdown mouse bone marrow mesenchymal stem cells (mBMSCs)

Mouse bone marrow mesenchymal stem cells (mBMSCs) were isolated from 6- to 8-week-old C57BL/6 male mice obtained from Shanghai Jihui Laboratory Animal Care Co., Ltd. Briefly, following antibiotic treatment, the femurs and tibias were dissected, and the bone marrow was flushed out using a sterile syringe. The cells were cultured in Alpha Minimum Essential Medium (α-MEM) (Hyclone, SH30265.01) supplemented with 10% fetal bovine serum (Gibco) and 1% penicillin-streptomycin (Gibco) for 24 h, and non-adherent cells were removed. The culture medium was replaced every 3 days, and cells were passaged with trypsin (Gibco) once confluence reached 80–90%. All experimental protocals were approved by the Animal Care and Use Committee of Shanghai Ninth People's Hospital, Shanghai JiaoTong University School of Medicine (SH9H−2024-A392-SB), in accordance with the Animals (Scientific Procedures) Act, 1986 (UK) (amended 2013).

For Rmp64 gene knockdown, shRNA lentiviral plasmids were constructed, and mBMSCs at passage 2 were transfected with either the knockdown or non-targeting control lentivirus. The lentiviral vectors contained an *EGFP* reporter gene and a puromycin resistance gene for positive selection. Cells were seeded in 10-cm culture dishes at a density of $1 \times 10^7$ cells per dish and subsequently infected with shRNA lentivirus in the presence of $8 \, \mu g \, mL^{-1}$ polybrene (Sigma Aldrich) for 24 h. The selection was performed using $4 \, \mu g \, mL^{-1}$ puromycin (Beyotime, ST551) for 48 h, and successful transfection was confirmed by fluorescence.

### Chondrogenic induction

Chondrogenic differentiation of mBMSCs was performed according to the protocol provided by the differentiation kit (Oricell, MUXMX-90041). For 3D spheroid differentiation, $4 \times 10^5$ mBMSCs were seeded into 15-mL centrifuge tubes and cultured in a chondrogenic induction medium at 37 °C for 21 days. For 2D differentiation, mBMSCs were seeded in 12-well plates at a density of $1 \times 10^5$ cells per well, with the differentiation medium replaced every 48–72 h.

### Alcian blue staining

Both 2D-cultured chondrogenic cells and 3D-cultured cartilaginous spheroids were washed twice with PBS and fixed in 4% paraformaldehyde (PFA) at room temperature for 30 min. The spheroids were subsequently embedded in Tissue-Tek O.C.T. Compound (SAKURA, 4583) and cryosectioned at a thickness of 10 μm. The cryosections, as well as the 2D-cultured chondrogenic cells, were stained with alcian blue solution (1% alcian blue in 0.1 M HCl, pH 1.0; Sigma Aldrich), followed by a wash in 0.1 M HCl and two washes with PBS[73].

### Proliferation assay

HEK293T cells were seeded at a density of 5000 cells per well into 96-well culture plates (Corning, 3904). The plates were then placed into the incucyte Live-Cell Analysis System and scanned every 4 h with a 10× objective. Live cell imaging analysis was performed with Incucyte Software utilizing the Cell-by-Cell Adherent analysis. Data from every well was normalized to the corresponding cell counts at the initial time point (0 h).

### Establishment of a stable cell line for purification

To isolate the endogenous human RNase MRP complex, we generated a stable cell line using a lentiviral transfection system to overexpress two protein components of RNase MRP. Specifically, the RNase MRP-specific protein RMP64 was cloned into a modified pLVX-IRES-EGFP lentiviral expression vector with a C-terminal twin-strep tag, and POP5 was cloned into a modified pLVX-IRES-mCherry vector with a C-terminal triple-Flag tag. These lentiviral constructs, together with the packaging plasmids, were transfected into HEK293T cells using X-tremeGENE HP transfection reagent (Roche). 48 h post-transfection, the virus-containing supernatant was collected and used to infect Expi293F cells (Thermo Fisher, A14527). After a 12-h incubation with fresh medium replacement, the cells were infected for an additional 2 days before being sorted by flow cytometry to isolate EGFP- and mCherry-double-positive cells, ensuring co-expression of both target proteins. HEK293T cells were cultured in DMEM supplemented with 10% FBS, while Expi293F cells were maintained in chemically defined Union-293 medium (Union-Bio, UP1000).

### Purification of human RNase MRP

For purification of human RNase MRP, 4 L of Expi293F cells overexpressing tagged RMP64 and POP5 proteins at a density of $4 \times 10^6$ cells/mL were harvested by centrifugation at $1000 \times g$ and washed once with PBS. The cell pellets were resuspended in 100 mL of lysis buffer containing 25 mM Hepes-K (pH 7.5), 150 mM NaCl, 10 mM MgCl$_2$, 1 mM DTT, 0.04% NP40, and a protease inhibitor cocktail. To specifically isolate RNase MRP, we used RMP64 (NEPRO) and POP5 as tandem-affinity purification baits in a two-step procedure. First, a Strep-tagged POP5 was used to enrich both the RNase P and RNase MRP complexes. Second, we employed Flag-tagged RMP64 (NEPRO), which is specific to RNase MRP, for affinity purification to ensure specificity. The purification protocol followed the same method previously described for the human RNase P complex[49].

### In vitro RNA cleavage assay

Pre-tRNA[Val] was in vitro transcribed by T7 polymerase using a synthetic DNA template, and purified by fast protein liquid chromatography (FPLC) (GE Healthcare). The ITS1 and ITS2 rRNA fragments were designed according to previously published data[4] (ITS1-A': 5'-UCCCGGGGGGCUCUUCGUGAUCGAUGUGGU-3', ITS1-3: 5'-CUGCG-GAAGGAUCAUUAACGGAGCCCGGAG-3', ITS1-E: 5'-CACCCACCCCCC-CACCGCGACGCGGCGCGU-3', ITS1-2: 5'-ACCCCUCUCCGGAGUCCGG UCC*CGUUUGCU-3', ITS1−2': 5'-GAGUCCGGUCC*CGUUUGCUGU-CUCGUCU-3', ITS2-1: 5'-GCCGCGCGCUCUCUCUCCCGUCGCCU-CUCCCCCUCGCCGGGCCC-3', *: cleavage site), chemically synthesized, and 5' labeled with FAM (5-Carboxyfluorescein) (Azenta Life Sciences). We mixed 2 μg of final purified pre-tRNA substrate and 50 ng of ITS1 rRNA substrate with the purified RNase P and MRP complexes at 37 °C for 30 min in 10-μl reaction buffer containing 25 mM Hepes-K pH 7.5, 150 mM NaCl, 1 mM DTT, 10 mM MgCl$_2$. The reactions were stopped by the addition of the loading buffer and denatured at 95 °C for 3 min. Samples were separated on a 10% urea-denaturing polyacrylamide gel, except for ITS1-2', which was separated on a 20% urea-denaturing gel. The gel of pre-tRNA was stained with 0.01% Ethidium Bromide. The 5'-FAM labeled ITS1 rRNA products were visualized by a Typhoon imaging system (GE Healthcare).

### Electron microscopy and image processing

Approximately 3 μL of the purified human RNase MRP complex was applied onto glow-discharged lacey-carbon grids (Ted Pella, 01824 G) using the PELCO easiGlow™ system. The grids were blotted for 3 seconds at 8 °C, plunged into liquid ethane using an FEI Vitroblast Mark IV at 100% humidity, and transferred to a Titan Krios transmission electron microscope operating at 300 kV. Cryo-EM images were recorded in super-resolution mode on a K3 camera (Gatan) at a nominal magnification of 81,000×, resulting in a pixel size of 0.55 Å and a dose rate of 17.9 e$^-$/Å$^2$ per second. The defocus range was set between −1.5 and −2.2 μm, with automated data collection using EPU software (FEI). Each image was recorded as 32 frames over a total exposure of 2.7 seconds

(cumulative dose ~50 e$^-$/Å$^2$). In total, 8332 micrographs were collected, motion-corrected, and dose-weighted using MOTIONCOR2[74], followed by CTF estimation with Gctf[75]. All further processing was performed using RELION4.0[76,77].

Approximately 1000 particles were manually picked and subjected to reference-free 2D classification to generate initial class averages, which were then used as templates for automated particle picking with Gautomatch (https://www.mrc-lmb.cam.ac.uk/kzhang/Gautomatch/), yielding a total of 6,307,025 particles. After several rounds of 2D classification, a refined dataset of 3,828,791 particles was obtained and used for 3D classification with a 20-Å low-pass filtered 200 kV cryo-EM map as the reference. The 3D classification revealed the typical features of the complex, notably a well-defined density for the large lobe. The selected particle subsets were then combined and refined via 3D refinement.

To further improve the resolution, the particles were subjected to several rounds of skip-alignment 3D classification. The best-quality major class was refined using a smaller mask focused on the top region of the complex in subsequent 3D auto-refinements. After particle polishing (beam-induced motion correction and radiation-damage weighting) and per-particle CTF refinement, 777,423 particles were used in a final 3D refinement, yielding a resolution of 2.8 Å (gold-standard FSC 0.143).

For the overall structure, a different 3D classification strategy was employed by increasing the T value. The major classes underwent two rounds of 2D classification and were selected for 3D refinement with recenter extraction. A total of 432,377 particles were auto-refined with an overall soft mask, producing a reconstruction at a 3.5-Å resolution (FSC 0.143). To improve the density of the small lobe, these particles underwent further skip-alignment 3D classification and were divided into two classes. The major class was re-extracted, and refined with various mask sizes. We employed a signal-subtraction workflow to remove the large-lobe contribution, followed by 3D classification focused on the small lobe. Finally, 294,518 particles were selected for 3D auto-refinement with an overall mask, resulting in a final resolution of 3.9 Å (FSC 0.143). All density maps were sharpened using a negative temperature factor determined automatically by the RELION4.0 post-processing program, and local resolution variations were also estimated using RELION4.0[76,77].

### Model building and refinement

An atomic model of the human RNase MRP complex was generated by combining de novo model building with rigid-body docking of components whose structures were previously determined. Model building was performed in COOT[78]. The three coaxially stacked RNA stems of the RMRP RNA were modeled based on the Nme1 core structure of yeast RNase MRP, followed by de novo modeling of the small lobe. Initially, known structures of the shared protein subunits from RNase P and MRP were docked into the EM density map. After assigning all RNA elements, the structures of the human RPP20, POP5-(RPP30)$_2$-RPP14-RPP40 heteropentamer, and the RPP29-RPP38 complex were fitted into the density. Our MS analysis confirmed that both RPP25 and RPP25L can be co-purified with RNase MRP. However, in the cryo-EM map, the corresponding density—located at the periphery of the complex—exhibits limited resolution. Regions of the map that could potentially distinguish between the two proteins are unclear. This ambiguity is further underscored by the high sequence similarity between RPP25 and RPP25L. Since the RPP25 model fits the density well without structural conflicts, we have used RPP25 in the RNase MRP structure. RMP64 and RMP24 were manually built based on the predictions from the AlphaFold database[70,71], and the N-terminal motifs of POP1 and RPP29 were also de novo constructed using structural information from yeast RNase MRP. Subsequent adjustments were made in COOT[78], and the final model was refined using phenix.real_space_refine with secondary structure and geometric restraints to avoid over-fitting[79]. Validation of the structure was performed using MolProbity[80] (see Supplementary Table 1), and all structural figures were generated with PyMol[81] or UCSF Chimera[82].

### Western blotting

Total proteins were extracted from $1 \times 10^6$ cells using RIPA buffer (Beyotime, #P0013B). Proteins were separated by SDS-PAGE and transferred onto PVDF membranes (Millipore, # IPVH00010). The membranes were blocked with a solution containing 5% BSA in PBS buffer supplemented with 0.05% TWEEN-20 at room temperature for 1 hr, and then incubated with anti-Flag antibody (HUABIO, #0912-1) at dilution of 1:5,000 and anti-Tubulin antibody (Proteintech, 14555-1-AP) at dilution of 1:20,000 in blocking buffer at 4 °C overnight. After washing, the membranes were incubated with HRP-conjugated secondary antibodies at room temperature for 1 hr. Following additional washes, the blots were developed using the ECL Prime Western Blotting System (GE Healthcare, RPN2232).

### In vivo pre-tRNA analysis

Total RNA was extracted from ~$2 \times 10^6$ cells expressing either mutant or wild-type RNAs using TRIzol reagent (Invitrogen). To quantify pre-tRNAs that contain a 5′-leader sequence, specific qPCR primers were designed: an F2 (forward) primer targeting the 5′-leader region of pre-tRNAs, and F1 (forward) and R (reverse) primers complementary to the mature tRNA region. By comparing the ratio of pre-tRNA to mature tRNA between mutant and wild-type RNA-expressing cells, we assessed the relative accumulation of the 5′-leader sequence. The primers were synthesis by Sangon Biotech and listed in Supplementary Data 5.

### Statistical analyses

The statistical details are indicated in the figure legends. Statistical analyses were performed using GraphPad Prism (v10.3.1) software. Two-tailed student's t-test and One-way ANOVA were used to evaluate the significance between groups. $P < 0.05$ was considered statistical significant. Experiments shown are representative of at least three replicates.

### Reporting summary

Further information on research design is available in the Nature Portfolio Reporting Summary linked to this article.

## Data availability

The accession numbers for the structure reported in this paper are PDB: 9UH7 (Large_lobe), 9UH9 (Holoenzyme), 9UHA (Small_lobe) and EMDB: EMD-64157 (Large_lobe), EMD-64159 (Holoenzyme), EMD-64160 (Small_lobe). Any additional information required in this paper is available upon request. Source data are provided with this paper.

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

## Acknowledgements

We thank the staff members of the Electron Microscopy System at Shanghai Institute of Precision Medicine for providing technical support and assistance in data collection. We thank Hong Lu, Shufang He, Jie Huang, Ying Cui, Rijing Liao, and Yan Cai from Shanghai Jiao Tong University School of Medicine (SHSMU) for help with confocal microscopy and mass spectrometry analyses. This work was supported by grants from the National Key Research and Development Program of China (2020YFA0509800 to P.L.), the National Natural Science Foundation of China (32471344 to P.L., 32471339 to J.W., 31930063 and 32430033 to M.L., 82072638 to Y.Z., 32400499 to X.W.), the Shanghai Municipal Education Commission Gaofeng Clinical Medicine Grant Support (20181711 to J.W.), the Innovative Research Team of High-level Local University in Shanghai (SHSMU-ZLCX20211700 to J.W. and M.L.), the National Research Center for Translational Medicine at Shanghai, Ruijin Hospital, Shanghai Jiao Tong University School of Medicine, Shanghai, China (NRCTM (SH) –2021-01 to M.L.) and the Taishan Scholars Program of Shandong Province (No. tsqn202408157 to F.W.). Ming Lei is a SANS Exploration Scholar.

## Author contributions

M.L., P.L., and B.Z. designed the study. M.L., P.L., J.W., and Y.J.Z. supervised the research. B.Z., X.W., and F.W. carried out the bulk of the experiments. B.Z., M.T., M.C., Y.S., and S.L. prepared cryo-EM specimens, collected data sets and determined the structure. J.W. carried out model building and refinement. B.Z., X.W., F.W., Y.Y.Z., X.Z., and R.G. carried out biochemical and evolutionary analysis. M.L., P.L., B.Z., X.W., and F.W. wrote the manuscript, with input from all authors. All authors edited and approved the final manuscript.

## Competing interests

The authors declare no competing interests.
