## [Transparent Peer Review file · Nature Communications]

Structural and Evolutionary Insights into the Eukaryotic RNase MRP Ribonucleoprotein Complex

Corresponding Author: Dr Ming Lei

Version 0:

Reviewer comments:

Reviewer #1

(Remarks to the Author)

I did review the first submission of this article. I have now read in detail the point by point answer to issues raised by all reviewers. The authors performed an extremely thorough revision work and answered in a perfect fashion absolutely all points raised by the reviewers. This is an excellent piece of work that I have no hesitation to recommend for publication in its present state.

Reviewer #2

(Remarks to the Author)

I was previously Reviewer 2 on this excellent report describing the first structure of human RNase MRP. All of my earlier concerns have been comprehensively addressed in the authors' responses and in the revised manuscript.

I have only three minor comments regarding the updated version:

(1) Statistical tests - there are several instances where multiple Student's t-tests are displayed on the same panel:
- in Figure 4E and 4H: would ANOVA be more appropriate here?
- in Extended Data Figure 20 panels A, C, D, E, F: would correction for multiple comparisons be appropriate here?

(2) Extended Data Fig 5D caption, line 91 - the caption currently reads:

"RMP64 knockdown impairs pre-rRNA processing in HEK293T cells. qRT-PCR analysis showing increased pre-ITS1/5' ETS ratio in RMP64-knockdown cells."

The term "pre-ITS1" is not standard and could be confusing. I suggest using "pre-cleaved ITS1," consistent with the terminology used in the main text.

(3) Reviewer Figure 7, panel B: this panel shows that knockdown of the shared MRP/P subunit POP1 results in a much more profound growth phenotype than knockdown of the newly identified MRP-specific subunits. This is an interesting finding and complements recent data from Liu. et al (PMID: 40533478) comparing the effect on growth of depletion of RNase P alone vs. both complexes. It would strengthen the manuscript if this result were included in the final version (e.g., as an Extended Data figure, rather than remaining solely in the rebuttal document).

Review by Nic Robertson, post-doctoral researcher, University of Edinburgh

Reviewer #3

(Remarks to the Author)

The authors have addressed most of my comments.

I have one remaining concern regarding the structural data and the associated models and figures. Upon inspection of the maps, model and validation files, it becomes evident that the stereochemistry of some parts of the models is not optimal. In particular, in all three models chain B has much poorer backbone stereochemistry than the remaining chains. More importantly, all three models show quite poor statistics regarding sidechain stereochemistry. Finally, some parts of the models show very poor model to map correlation, which is also reflected in Q scores. An example for this is chain A, residues 127-146 in 9UH9 (holoenzyme), which have very poor density and could just as well be left out of the final model in my opinion (including the opposing strand of RNA). The poor Q score is particularly pronounced for the small lobe (which is somewhat expected, given that this has the poorest map quality). Taken together, I would advise the authors to revise their models and pursue a more conservative modelling and refinement strategy, for example with imposing tighter restraints on stereochemistry and leaving out poorly resolved parts in the final model, prior to final PDB deposition.

In Ext. Data Fig. 7, I think the FSC plots are incorrectly labelled, as it does not make sense that the masked map reaches lower resolution than unmasked. I believe the red line must be the phase randomized FSC, the blue line the masked Map and the green the phase randomization corrected FSC. I would ask the author to double-check this figure. Also, final numbers of particles for each map should be given in the figure.

Version 1:

Reviewer comments:

Reviewer #2

(Remarks to the Author)

All my points on this revised manuscript have been comprehensively addressed with further revisions to the text and figures, and I support publication. Thank you for the opportunity to review this interesting work.

Reviewer #3

(Remarks to the Author)

The authors have addressed my comments and improved the structural models.

Point-to-point response to reviewers' comments

**(Structural and Evolutionary Insights into the Eukaryotic RNase MRP
Ribonucleoprotein Complex; NCOMMS-25-71390-T)**

Reviewer #1

I did review the first submission of this article. I have now read in detail the point-by-point answer to issues raised by all reviewers. The authors performed an extremely thorough revision work and answered in a perfect fashion absolutely all points raised by the reviewers. This is an excellent piece of work that I have no hesitation to recommend for publication in its present state.

Thanks!

Reviewer #2

I was previously Reviewer 2 on this excellent report describing the first structure of human RNase MRP. All of my earlier concerns have been comprehensively addressed in the authors' responses and in the revised manuscript.

I have only three minor comments regarding the updated version:

(1) Statistical tests -there are several instances where multiple Student's t-tests are displayed on the same panel:

- in Figure 4E and 4H: would ANOVA be more appropriate here?
- in Extended Data Figure 20 panels A, C, D, E, F: would correction for multiple comparisons be appropriate here?

Thanks for this good point. Following the reviewer's suggestion, we have now re-analyzed the data for Figure 4E, 4H and Extended Data Figure 20 (panels A, C, D, E, F) using a One-way ANOVA for multiple comparisons. The figures (Reviewer Figures 1 and 2) and their legends have been updated accordingly.

Reviewer Figure 1. (Revised Figure 4 e and h) Immunofluorescence analysis of pre-rRNA processing defects caused by the alanine substitution of His146^{POP1} (a) and disease-related mutations in RMP64 or POP1 (b). For all experiments, data were shown as mean ± SD from three independent replicates. Significance was determined using One-way ANOVA. *p < 0.001, ****p < 0.0001.**

Reviewer Figure 2. (Revised Extended Data Figure 20) Functional analysis of human RNase MRP. **a**, Quantitative RT-PCR analysis of relative expression levels of overexpressed WT and mutant RPP29. **b**, Western blot analysis of Flag-tagged WT and mutant RPP29. **c**, Quantitative RT-PCR analysis revealed elevated pre-cleaved ITS1 levels relative to the 5'-external transcribed spacer (5' ETS) in RPP29 mutant-expressing cells compared to control cells. Both pre-cleaved ITS1 and the 5' ETS signals were normalized to 18S. **d**, Quantitative RT-PCR analysis of the expression levels of wild-type (WT) or the H146A mutant of POP1 in control cells, and in POP1-knockdown cells expressing empty vector, WT POP1, or the H146A mutant of POP1. **e**, Quantitative RT-PCR analysis of pre-cleaved ITS1 levels relative to the 5'-external transcribed spacer (5'-ETS) in control cells, and in POP1-knockdown cells expressing empty vector, WT POP1, or the H146A mutant of POP1. **f**, Quantitative RT-PCR analysis of the expression levels of WT or disease-causing mutants of RMP64 and POP1 in control cells, in RMP64-knockdown cells expressing empty vector, WT RMP64, or the R94C and L145F mutants of RMP64, and in POP1-knockdown cells expressing empty vector, WT POP1, or the P582S and G583E mutants of POP1. For all experiments, data were shown as mean \pm SD from three independent replicates. Significance was determined using One-way ANOVA. * $p < 0.05$, ** $p < 0.01$, *** $p < 0.001$, **** $p < 0.0001$.

(2) Extended Data Fig 5D caption, line 91 - the caption currently reads:

“RMP64 knockdown impairs pre-rRNA processing in HEK293T cells. qRT-PCR analysis showing increased pre-ITS1/5' ETS ratio in RMP64-knockdown cells.

The term “pre-ITS1” is not standard and could be confusing. I suggest using “pre-cleaved ITS1”, consistent with the terminology used in the main text.

Thanks. As recommended, we have now used the term “pre-cleaved ITS1” in the revised manuscript:

“d, RMP64 knockdown impairs pre-rRNA processing in HEK293T cells. qRT-PCR

analysis showing increased pre-cleaved ITS1/5' ETS ratio in RMP64-knockdown cells.”
(Supplementary information, page 10)

(3) Reviewer Figure 7, panel B: this panel shows that knockdown of the shared MRP/P subunit POP1 results in a much more profound growth phenotype than knockdown of the newly identified MRP-specific subunits. This is an interesting finding and complements recent data from Liu et al (PMID: 40533478) comparing the effect on growth of depletion of RNase P one vs. both complexes. It would strengthen the manuscript if this result were included in the final version (e.g., as an Extended Data figure, rather than remaining solely in the rebuttal document).

Thanks for this good point. Following the reviewer’s suggestion, we have added the growth phenotype data for POP1-knockdown cells to the revised manuscript (now in Extended Data Figure 6g) and have made the following corresponding changes:

“Furthermore, depletion of these subunits resulted in pronounced growth defects characterized by reduced proliferation without loss of viability (Extended Data Figs. 5g, 6f). The more severe phenotype observed upon POP1 knockdown compared with RMP64 or RMP24 depletion underscores that concurrent impairment of RNase P and MRP more strongly affects cellular growth (Extended Data Fig. 6g)⁴⁴.” (Page 8)

[44] Liu, Y., He, S., Pyo, K. *et al.* Reversible proliferative arrest induced by rapid depletion of RNase MRP. *Nat Commun* 16, 5342, doi: 10.1038/s41467-025-60471-4 (2025).

Reviewer Figure 3. (Revised Extended Data Figure 6g) Proliferation analysis of control and POP1 knockdown cells. Images were acquired every 4 hours using the Incucyte software, and analyzed by the Cell-by-Cell adherent analysis. For all experiments, data were shown as mean \pm SD from at least three independent replicates. The statistical significance at the final timepoint of the growth curve (g) was calculated using One-way ANOVA. **** $p < 0.0001$.

Reviewer #3

The authors have addressed most of my comments. I have one remaining concern regarding the structural data and the associated models and figures. Upon inspection of the maps, model and validation files, it becomes evident that the stereochemistry of some parts of the models is not optimal. In particular, in all three models chain B has much poorer backbone stereochemistry than the remaining chains. More importantly, all three models show quite poor statistics regarding sidechain stereochemistry. Finally, some parts of the models show very poor model to map correlation, which is also reflected in Q scores. An example for this is chain A, residues 127-146 in 9UH9 (holoenzyme), which have very poor density and could just as well be left out of the final model in my opinion (including the opposing strand of RNA). The poor Q score is particularly pronounced for the small lobe (which is somewhat expected, given that this has the poorest map quality). Taken together, I would advise the authors to revise

their models and pursue a more conservative modelling and refinement strategy, for example with imposing tighter restraints on stereochemistry and leaving out poorly resolved parts in the final model, prior to final PDB deposition.

We thank the reviewer for the constructive suggestion on structural improvement. Following the reviewer's suggestion, we undertook a revision of all three structural models.

As suggested, we have removed the poorly defined regions (Chain A, nucleotides 131-146 in 9UH9) from the holoenzyme model. Subsequently, we performed rigorous manual optimization of the backbone and side-chain geometries for all three models. These refinements resulted in an improved stereochemical quality of the models, as validated by wwPDB and RAMPAGE statistics (Reviewer Figures 4, 5 and 6). The final models were further refined through iterative real-space refinement, which improved their fit to the electron density maps and yielded better Q scores (Reviewer Figure 7).

[REDACTED]

Reviewer Figure 4. MolProbity analysis of backbones. a-b, Protein backbone analysis of human RNase MRP large lobe before **(a)** and after optimization **(b)**. **c-d,** Protein backbone analysis of human RNase MRP holoenzyme before **(c)** and after optimization **(d)**. **e-f,** Protein backbone analysis of human RNase MRP small lobe before **(e)** and after optimization **(f)**. The Percentiles column shows the percent Ramachandran outliers of the chain as a percentile score with respect to all PDB entries followed by that with respect to all EM entries. The Analysed column shows the number of residues for which the backbone conformation was analysed, and the total number of residues.

[REDACTED]

Reviewer Figure 5. MolProbity analysis of sidechains. a-b, Protein sidechain analysis of human RNase MRP large lobe before (**a**) and after optimization (**b**). **c-d,** Protein sidechain analysis of human RNase MRP holoenzyme before (**c**) and after optimization (**d**). **e-f,** Protein sidechain analysis of human RNase MRP small lobe before (**e**) and after optimization (**f**). The Percentiles column shows the percent sidechain outliers of the chain as a percentile score with respect to all PDB entries followed by that with respect to all EM entries. The Analysed column shows the number of residues for which the sidechain conformation was analysed, and the total number of residues.

Reviewer Figure 6. Ramachandran plot analysis of the refined atomic models of human RNase MRP large lobe (a), holoenzyme (b), and small lobe (c).^[1]

[1] S.C. Lovell, I.W. Davis, W.B. Arendall III, P.I.W. de Bakker, J.M. Word, M.G. Prisant, J.S. Richardson & D.C. Richardson. Structure validation by $C\alpha$ geometry: ϕ/ψ and $C\beta$ deviation. *Proteins: Structure, Function & Genetics*, 50: 437-450 (2002)

Reviewer Figure 7. Map-model fit summary. **a-b**, the atom inclusion and Q-score of human RNase MRP large lobe before (**a**) and after optimization (**b**). **c-d**, the atom inclusion and Q-score of human RNase MRP holoenzyme before (**c**) and after optimization (**d**). **e-f**, the atom inclusion and Q-score of human RNase MRP small lobe before (**e**) and after optimization (**f**). The average atom inclusion at the recommended contour level (0.003) and Q-score for the entire model and for each chain.

In Ext. Data Fig.7, I think the FSC plots are incorrectly labelled, as it does not make sense that the masked map reaches lower resolution than unmasked. I believe the red line must be the phase randomized FSC, the blue line the masked Map and the green the phase randomization corrected FSC. I would ask the author to double-check this figure. Also, final numbers of particles for each map should be given in the figure.

Thanks for pointing out this mistake, we have now corrected this error in the revised Extended Data Figure 7 (Reviewer Figure 8).

Furthermore, as suggested, we have now included the final numbers of particles used to generate each map in the Extended Data Figure 7.

Reviewer Figure 8. (Revised Extended Data Figure 7 e, i and m) Fourier Shell Correlation curves of the reconstructed maps of the RNase MRP large lobe (a), small lobe (b), and RNase MRP holoenzyme complex (c).